# RSV glycoprotein and genomic RNA dynamics reveal filament assembly prior to the plasma membrane

Daryll Vanover[1], Daisy V. Smith[1], Emmeline L. Blanchard[1], Eric Alonas[1], Jonathan L. Kirschman[1], Aaron W. Lifland[2], Chiara Zurla[1] & Philip J. Santangelo[1]

The human respiratory syncytial virus G protein plays an important role in the entry and assembly of filamentous virions. Here, we report the use of fluorescently labeled soybean agglutinin to selectively label the respiratory syncytial virus G protein in living cells without disrupting respiratory syncytial virus infectivity or filament formation and allowing for interrogations of respiratory syncytial virus virion assembly. Using this approach, we discovered that plasma membrane-bound respiratory syncytial virus G rapidly recycles from the membrane via clathrin-mediated endocytosis. This event is then followed by the dynamic formation of filamentous and branched respiratory syncytial virus particles, and assembly with genomic ribonucleoproteins and caveolae-associated vesicles prior to re-insertion into the plasma membrane. We demonstrate that these processes are halted by the disruption of microtubules and inhibition of molecular motors. Collectively, our results show that for respiratory syncytial virus assembly, viral filaments are produced and loaded with genomic RNA prior to insertion into the plasma membrane.

[1] Wallace H. Coulter Department of Biomedical Engineering, Georgia Institute of Technology and Emory University, Atlanta, Georgia 30332, USA. [2] Institute of Bioengineering and Bioscience, Georgia Institute of Technology, Atlanta, Georgia 30332, USA. Correspondence and requests for materials should be addressed to P.J.S. (email: philip.santangelo@bme.gatech.edu)

Respiratory syncytial virus (RSV) remains the leading cause of acute lower respiratory infections worldwide in children under 5 years of age and leads to roughly 3 million hospital admissions each year[1]. Despite the high global incidence rate of infected patients, no effective vaccine yet exists[2]. While various treatments are being actively investigated, the cellular events that occur during RSV assembly are poorly understood.

RSV is a member of the *Paramyxoviridae* family and contains a single-stranded negative-sense RNA genome that encodes for 11 proteins. The RSV fusion protein (RSV F) is one of three encoded glycoproteins and is necessary for viral fusion with the cellular membrane and subsequent entry into host cells. Once virions are intracellular, the large (RSV L) polymerase, working with the RSV nucleoprotein (RSV N), and phosphoprotein (RSV P), transcribes viral genomic RNA into messenger RNA (mRNA), which encode viral proteins. Viral transmembrane proteins are post-translationally glycosylated and transported by the secretory membrane system to the plasma membrane, where they interact with ribonucleoprotein (RNP) complexes consisting of RSV N, RSV P, potentially RSV matrix protein (RSV M), a structural protein, and viral genomic RNA[3, 4]. The virus then assembles into mature pleomorphic particles which are either spherical or filamentous[5, 6]. Formation of these filaments requires RSV F, RSV N, RSV P, and RSV M, and has been found to contribute to higher viral titers, potentially through their contribution to cell to cell spreading of the virus[7–10]. Some reports hypothesized that RSV might utilize a second maturation pathway, where virions bud into intracellular vesicles forming filaments, but these findings have not been confirmed or further investigated[11, 12]. Several studies demonstrated that the RSV M dimerizes and binds to the cytoplasmic tail of RSV F, and thus plays a major role in the production of filamentous virions[13, 14]. RSV M also appears to be responsible for the maturation and elongation of RSV filaments[15]. Additionally, RSV M has been shown to assemble into filamentous structures in vitro[16]. Even though there has been significant emphasis on the roles of RSV F and M during filament formation, the steps leading to filament assembly are not clear.

A variety of host-cell factors, particularly cytoskeleton components, and viral proteins have been implicated in RSV filament formation, however our understanding of their mechanistic role is limited. Both ß-actin and actin-associated proteins were found in sucrose gradient-purified RSV preparations[17]. Additionally, actin was found to be primarily involved with virion egress, but has also been implicated in filament production[18, 19]. Indeed, inhibition of RhoA, an actin modulatory kinase, results in a shift to more spherical virion morphologies, and disruption of the actin network, which also leads to halting of RSV filament motion[8, 20]. Microtubules have also been shown to play a key role in the assembly of progeny RSV[18]. In contrast, other groups have suggested that filaments can form independently of the host-cell cytoskeleton[7].

The viral G protein is a highly glycosylated 90 kDa transmembrane protein, primarily responsible for the attachment of RSV to the host cell[21–23]. Though not required for the production of infectious RSV or virus-like particles, RSV G is necessary for full infectivity and is found on the membrane of mature filaments[12, 24, 25]. Additionally, RSV G interacts with both RSV M and F protein, which appears to be required for complete virion assembly[26, 27]. During initial virus entry, RSV glycoproteins have been shown to internalize via clathrin-mediated endocytosis, whereas caveolin is heavily incorporated into the envelope of mature RSV filaments[28–31]. RSV G was also found to be cleaved in the Vero cell line during endosomal recycling, and has been implicated in the maturation and egress of RSV[3, 5, 29, 32]. While the RSV G glycoprotein is found within progeny viral filaments and is clearly essential for virion entry into host cells, little data exists on the dynamics or trafficking of RSV G during infection, due to a lack of available reagents for tracking RSV G in live cells[33].

Previously, we have demonstrated the use of single-molecule sensitive RNA probes for imaging RSV viral genomic RNA in live cells, but the lack of a method for tracking the viral transmembrane proteins precluded our ability to image viral particle formation. In order to target the G protein, we investigated the use of the lectin soybean agglutinin (SBA) for specific labeling of RSV G, and found that SBA binding did not affect RSV infectivity, replication, or formation of viral filaments. Using fluorescently conjugated SBA, we report here that membrane-bound RSV G recycles from the plasma membrane via clathrin-mediated endocytosis into vesicles. We also show that these recycling endocytic vesicles merge with intracellular vesicles containing viral genomic RNPs and eventually associate with caveolin before returning to the plasma membrane. To the best of our knowledge, this is the first live imaging of a wild-type RSV viral protein (RSV G) and genomic RNA during RSV filament formation which revealed: the dynamics of filament formation, the production of branched filamentous structures typical of late-stage infections, the importance of microtubules in the assembly and loading of these filamentous structures in the cytosol, and the halting of this rapid motion once these structures merge with the plasma membrane. Together, these results change our understanding of RSV filament assembly and loading prior to the membrane and open new avenues for research and RSV therapeutics.

## Results

**Fluorescent SBA conjugates specifically bind RSV G**. RSV G was previously found to contain N-acetylgalactosamine (GalNAc) side chains, which interact specifically with various lectins[34]. We first analyzed the localization of Alexa Fluor 488 conjugated SBA (SBA-488) in cells infected with the A2 strain of human RSV with respect to RSV G and RSV F. Immediately prior to infection, SBA was delivered to cells at 4 °C, in order to minimize non-receptor mediated endocytosis. Upon infection for 24 h, both RSV G and RSV F colocalized with SBA-488, with minimal background in mock-infected controls (Fig. 1a). In addition, the SBA-488 signal in both the membrane and filaments was higher in infected cells than in control cells (Fig. 1b), which indicated that a large portion of glycoprotein was present in the membrane (Fig. 1a). Due to previous reports describing the binding of RSV F and RSV G in solution, we opted to examine a reduced system, using microscopy and expression of individual viral proteins, instead of an immunoprecipitation assay to demonstrate the specificity of SBA for RSV G[35]. Vero cells were transfected with in vitro transcribed (IVT), modified mRNA that encodes for either RSV F or RSV G[36]. We then labeled the cells with SBA-488 before fixing and immunostaining, without permeabilization, for RSV F and G (Fig. 1c). While transfection with RSV F did not alter background SBA staining, cells transfected with RSV G showed a marked increase in SBA staining relative to untransfected control cells, due to the heavy glycosylation of the RSV G protein. SBA colocalized in cells transfected with RSV G and not with RSV F, as indicated by both the Pearson's correlation coefficient and Mander's colocalization coefficient (Fig. 1d). A Pearson's correlation value much greater than 0 indicates a significant positive correlation between RSV G and SBA. The significantly higher Mander's coefficient indicates that a much larger percentage of the RSV G signal contains SBA signal, while very little of the RSV F signal contains SBA. The expressed RSV G and RSV F both localized to the plasma membrane, evidenced by the distribution of the viral proteins across the cell (Fig. 1c) and by the layer of

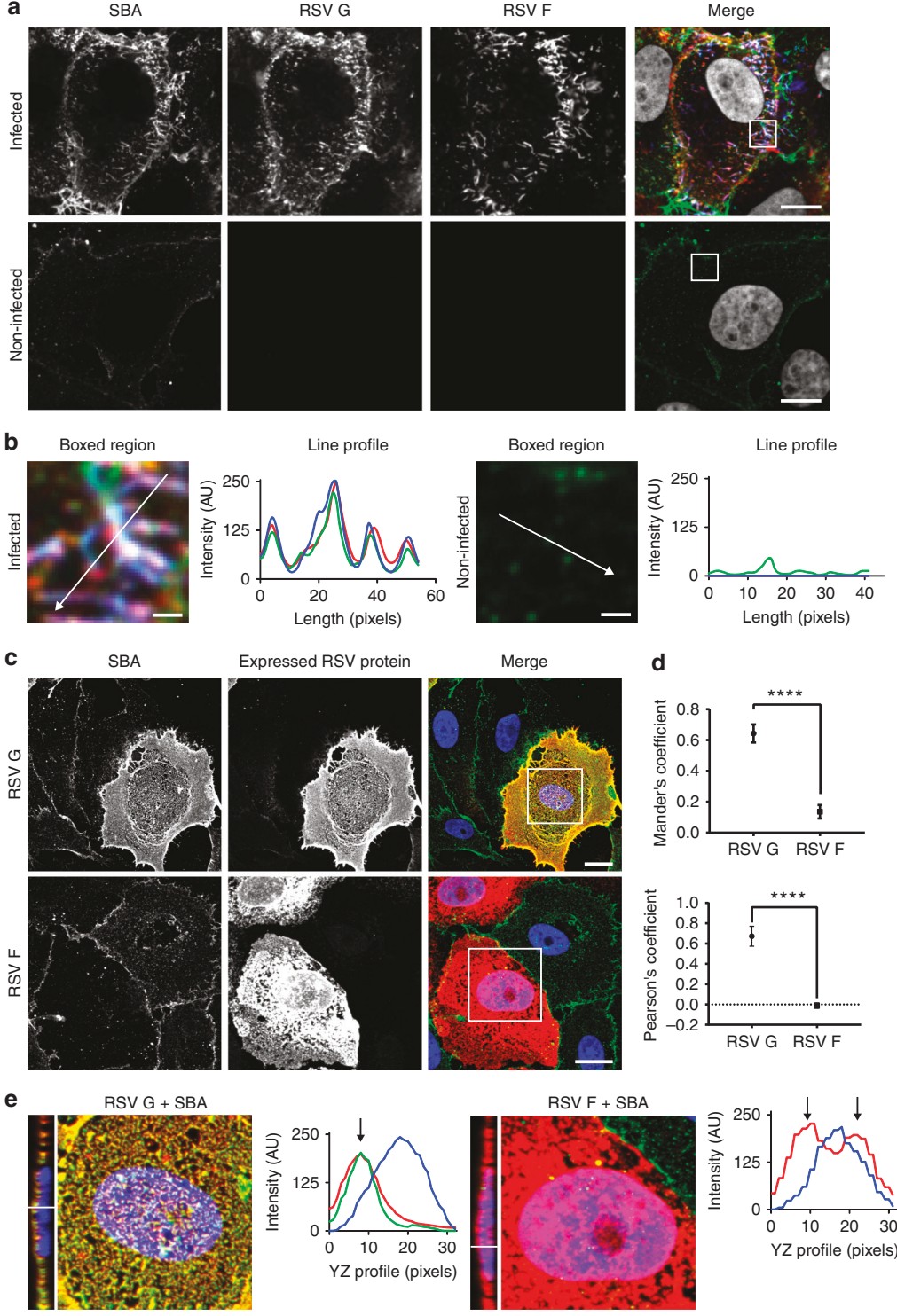

**Fig. 1** SBA selectively binds RSV G. **a** Vero cells infected (*top*) or mock-infected (*bottom*) for 24 h with RSV A2 at an MOI of 1 were labeled with SBA-488 at 4 °C (*green*) before being fixed and stained for RSV G (*red*), RSV F (*blue*), and nuclei (*gray*). Extended focus image is shown. *Scale bar* represents 10 μm. **b** Enlarged cropped images from the *white boxed regions* from part **a** with intensity profiles showing colocalization along the direction of the *white arrow*. *Scale bar* represents 1 μm. **c** Cells transfected for 24 h with RSV G (*top*) and RSV F (*bottom*) mRNA were labeled with SBA-488 at 4 °C (*green*) before being fixed and stained without permeabilization for RSV G or RSV F (*red*) and nuclei (*gray*). Extended focus image is shown. *Scale bar* represents 15 μm. **d** Mander's colocalization coefficient and Pearson's correlation between RSV G or RSV F and SBA. *Dotted line* indicates 0 correlation. *n* = 16 cells per group. *Error bars* represent 95% confidence interval. *Asterisks* indicates *p* < 0.0001 by Mann–Whitney *U* Test. **e** Enlarged cropped images from the *white boxed regions* from part **c** with YZ cross-sections (*left*). Intensity profiles, indicated by the *white line* on the YZ cross-section, show the expressed RSV protein on a higher plane than the nucleus as indicated by the *arrow*. Extended focus cropped images are shown

protein occupying a higher imaging plane than the nuclei of the cells (Fig. 1e). Overall, these results indicate that SBA can be used to specifically label RSV G, supporting previous lectin-binding studies to RSV glycoproteins[34].

**SBA conjugates do not inhibit replication of RSV.** We subsequently verified that SBA does not alter RSV replication or

infectivity. First, we labeled RSV A2 virus stocks with SBA-488. Then, we isolated the filamentous and spherical viral particles and deposited them onto glass coverslips as described previously[6]. Using fluorescence imaging and colocalization analysis, we verified that SBA was bound to both filamentous and spherical particles, which also contained RSV F and RSV N (Fig. 2a). Note that RSV infected cells generate incomplete and complete virions, as indicated by particles that contain only some of the various

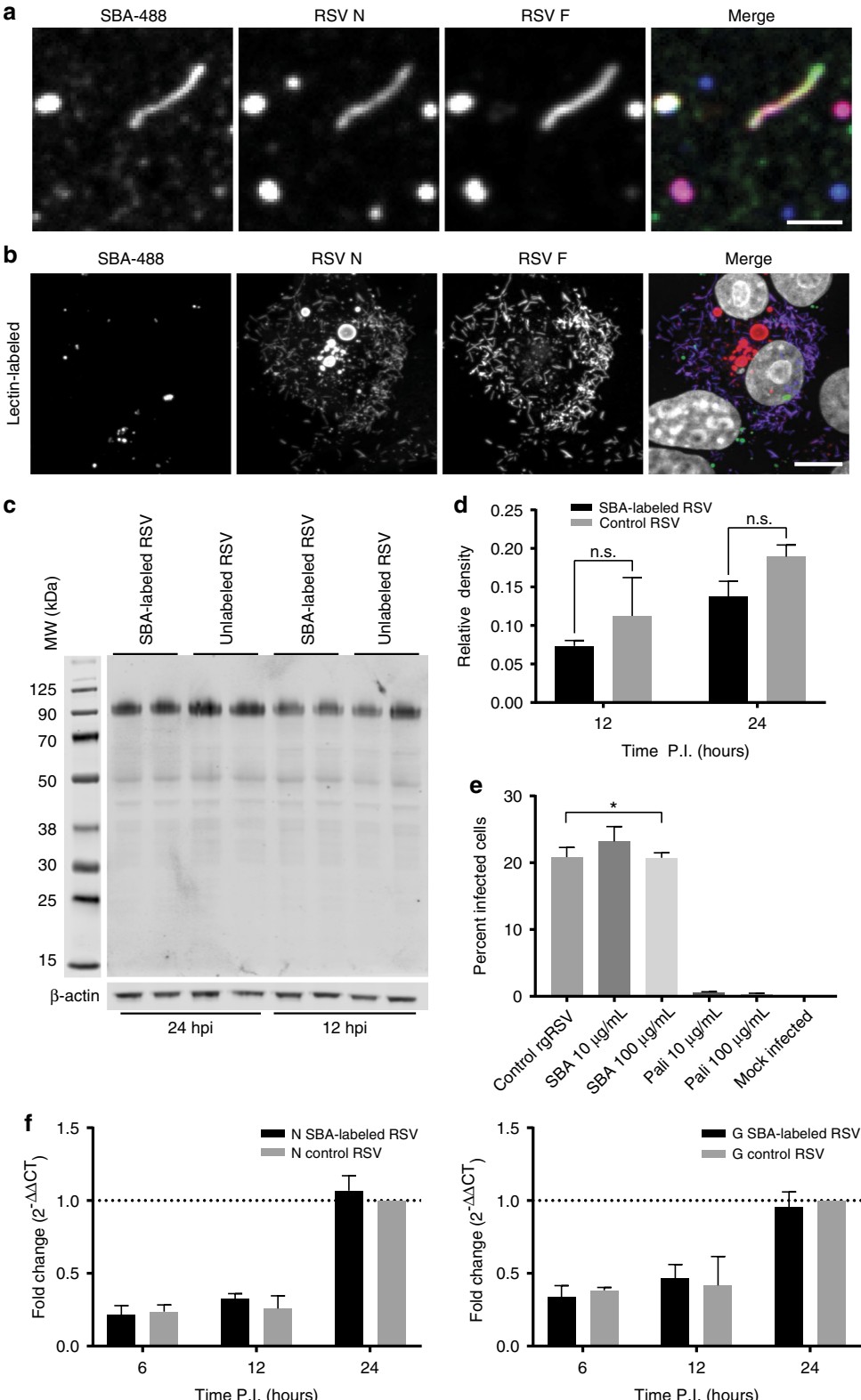

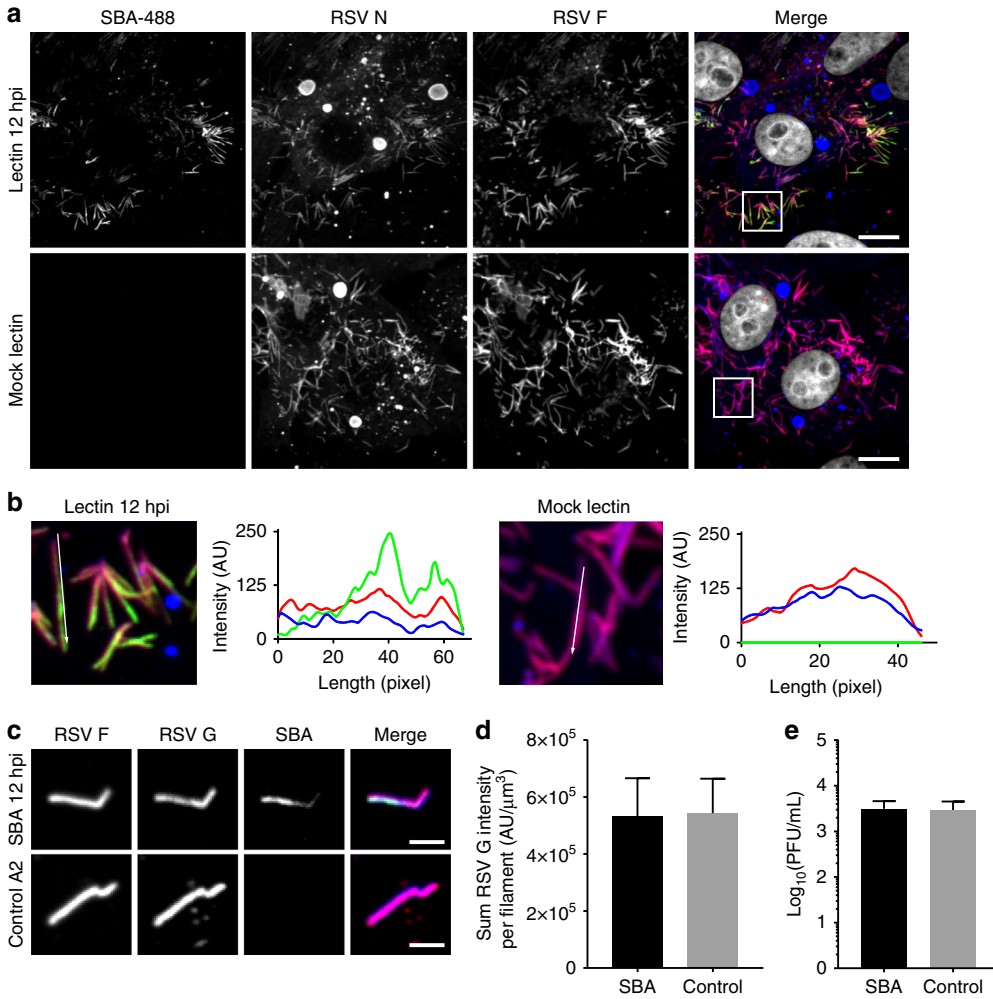

**Fig. 3** SBA allows for the visualization of RSV G dynamics in RSV infected cells. **a** Vero cells were infected with RSV A2 at an MOI of 1. At 12 hpi, cells were stained live with SBA-488 (*top*) or vehicle control (*bottom*), washed, and further incubated in complete growth media. 24 hpi, cells were fixed and immunostained for RSV N (*blue*), RSV F (*red*), and nuclei (*gray*). *Scale bar* represents 10 μm. **b** Enlarged cropped images from the *white boxed regions* from part **a** with intensity profiles showing colocalization along the direction of the *white arrow*. **c** Vero cells were infected with RSV A2 at an MOI of 1. 12 hpi, cells were stained live with SBA-488 (*top, green*) or vehicle control (*bottom*), washed, and further incubated in complete growth media. 24 hpi, virus was harvested. Filaments were isolated onto glass and stained for RSV F (*blue*) and RSV G (*red*) *Scale bar* represents 2 μm. **d** Mean sum of RSV G intensity per filament was quantified. 50 isolated virions were analyzed for each condition. *Error bars* represent standard deviation. There was no statistically significant difference between SBA-labeled and unlabeled RSV (Mann–Whitney *U* test, $p > 0.05$). **e** Vero cells were infected with RSV A2 at an MOI of 1. 12 hpi, cells were stained live with SBA-488 or vehicle control, washed, and further incubated in complete growth media. 24 hpi, supernatants from single infected wells were titered by plaque assay in quadruplicate. Three wells were independently infected, stained, and assayed per treatment. *Error bars* represent standard deviation. There was no statistically significant difference between SBA-labeled and unlabeled RSV (Mann–Whitney *U* test, $p > 0.05$)

**Fig. 2** SBA binding does not inhibit RSV binding or replication. **a** SBA-488 (*green*) was incubated with RSV A2 stocks for 30 min before filaments were isolated onto glass coverslips. Isolated virions were fixed and stained for RSV N (*red*) and RSV F (*blue*). *Scale bar* represents 2 μm. **b** SBA-488 labeled virus (*green*) was used to inoculate Vero cells at an MOI of 1. At 24 hpi, cells were fixed and immunostained for RSV N (*red*), RSV F (*blue*), and nuclei (*gray*). *Scale bar* represents 10 μm. **c** RSV A2 was incubated with either SBA or vehicle control for 2 h at 37 °C. Virus was then used to inoculate Vero cells at an MOI of 1. At 12 or 24 hpi, cells were lysed, and lysates were assayed by western blotting for RSV G. β-actin is included as a loading control. Image is representative of two independent experiments. **d** Replicate lanes ($n = 2$) were quantified by densitometry to compare the amount of RSV G signal from SBA-labeled or unlabeled RSV infections. There were no statistically significant differences at either time point (two-way ANOVA, $p > 0.05$). Error bars represent standard deviation. **e** rgRSV-GFP was incubated with the indicated concentrations of either palivizumab, SBA, or vehicle control for 2 h at 37 °C. Virus treatments and a mock treatment were then used to inoculate Vero cells at an MOI of 1. At 48 hpi cells were lifted into a single-cell suspension and analyzed by flow cytometry. Data represents the mean percentage of infected cells in three independent experiments. *Error bars* represent standard deviation. *Asterisk* indicates significant difference vs. mock infection (one-way ANOVA on ranks, $p < 0.05$). **f** Relative quantity of RSV N (*left*) and RSV G (*right*) expression via qRT-PCR upon infection of Vero cells with SBA-labeled or unlabeled RSV A2 at an MOI of 1 at 6, 12, and 24 hpi. There was no statistically significant difference at any time point (two-way ANOVA, $p > 0.05$). *Error bars* represent standard deviation. Results are the mean of two independent experiments normalized by the control virus at 24 hpi

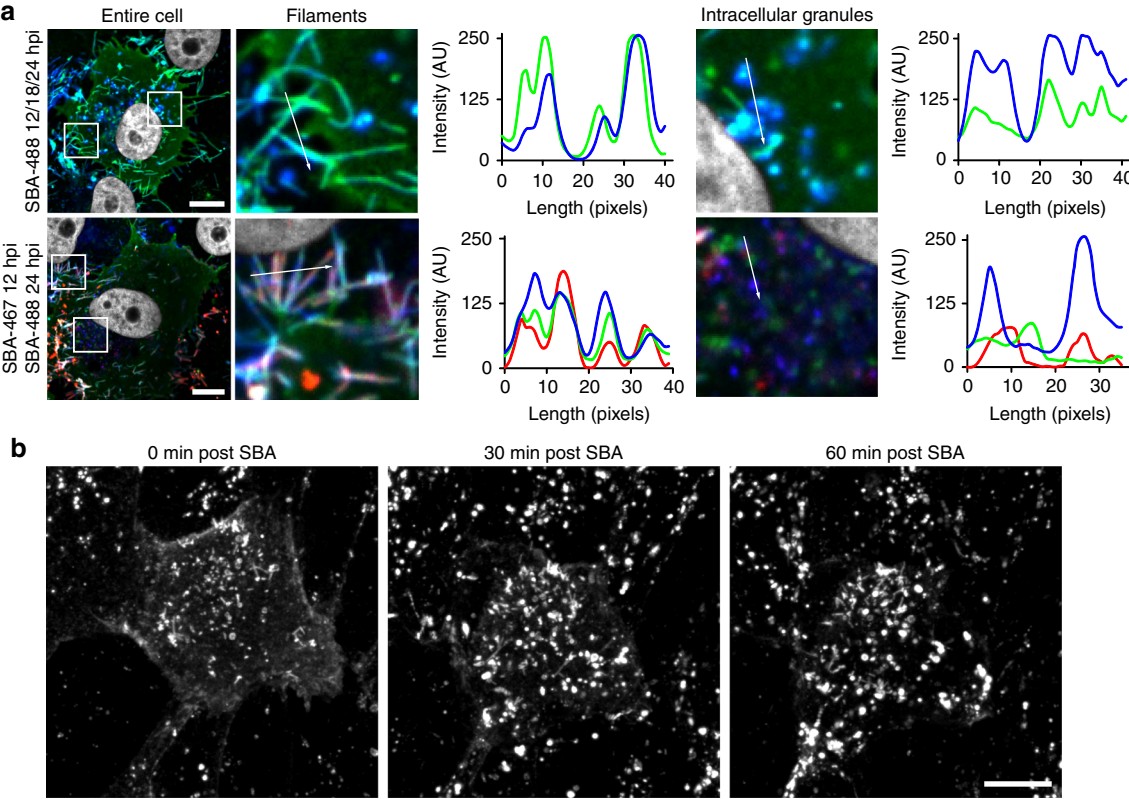

**Fig. 4** RSV G recycles from the plasma membrane into intracellular vesicles. **a** Vero cells were infected with RSV A2 at an MOI of 1. Cells were either labeled with SBA-488 (*green*) at 12, 18, and 24 hpi (*top*) or with SBA-647 (*red*) at 12 hpi and SBA-488 at 24 hpi. Cells were then fixed and stained for RSV F (*blue*). Enlargements of cropped images of either filaments or intracellular vesicles from the *white boxed regions* with intensity profiles showing colocalization along the direction of the *white arrow*. *Scale bar* represents 10 μm. **b** Vero cells were infected with RSV for 12 hpi before being labeled with SBA-488 (*green*). Immediately following SBA delivery, cells were imaged live over 1 h. Extended focus images shown. *Scale bars* represent 10 μm

viral proteins[6]. To demonstrate that labeling RSV A2 with SBA-488 did not significantly inhibit viral replication, we infected cells for 24 h and stained for RSV F and RSV N (Fig. 2b). Both cytosolic IBs and viral filaments were found, indicative of late-stage infections.

We then sought to determine the effect of SBA-labeled RSV A2 on virus infectivity. First, we bound SBA to RSV A2, similarly to assays described previously[37]. We then infected cells and measured RSV G protein synthesis relative to unlabeled RSV A2 after 12 and 24 h via western blot (Fig. 2c). Both the untreated and SBA-treated lysates contained similar amounts of RSV G at both time points. We quantified the amounts of protein using densitometry, and found that the differences at either time point were not statistically significant (Fig. 2d). To further test whether SBA affects RSV infectivity, we incubated RSV virus modified to express recombinant green fluorescent protein (rgRSV-GFP) with either palivizumab or SBA and measured GFP signal via flow cytometry[38, 39]. Because rgRSV-GFP has reduced permissivity compared to RSV A2, we allowed the infection to proceed for 48 h, before assaying the cells for GFP signal. SBA-labeled rgRSV-GFP or unlabeled virus infected a similar percentage of cells, while palivizumab-labeled virus abrogated any productive infection (Fig. 2e). Last, we compared via quantitative reverse transcription PCR (qRT-PCR) the levels of RSV N and G mRNA at 6, 12, and 24 h post-infection (hpi) with SBA-labeled RSV and unlabeled RSV, and found no statistically significant differences (Fig. 2f).

**SBA labels RSV G without inhibiting viral protein production.** Since SBA does not significantly impair the replication or

infectivity of RSV A2 when bound to virions prior to infection, we next sought to investigate if we could label RSV G on cells with SBA during an infection, without impairing viral processes. Because the majority of filaments are produced between 12 and 24 hpi, we incubated RSV infected cells with SBA-488 or vehicle control 12 h hpi and allowed the cells to incubate a further 12 h before fixation. Cells were then stained for RSV N and RSV F (Fig. 3a). Labeling of RSV G with SBA-488 12 hpi was specific, and especially stained viral filaments in infected cells with minimal background (Fig. 3b).

We then verified that labeling of RSV G with SBA-488 12 hpi in live cells was not altering the amount of RSV G present in filamentous virions. To do this, we stained live, RSV infected cells with SBA-488 or vehicle control at 12 hpi, and harvested the virus at 24 hpi. Then we isolated viral filaments and deposited them onto glass and immunostained for RSV F and RSV G (Fig. 3c). The amount of RSV G per filament was similar in SBA labeled and unlabeled RSV virions (Fig. 3d). Finally, we verified that labeling RSV infected cells with SBA did not inhibit the release of progeny virions. We stained RSV infected cells with either SBA or vehicle control at 12 hpi, isolated the supernatant at 24 hpi, and measured the titer by plaque assay (Fig. 3e). There was no statistically significant difference between viral titers from SBA labeled and unlabeled RSV infected cells. Overall these results indicated that SBA can be utilized to specifically label RSV G in live cells without altering viral protein expression and localization.

**RSV G recycles from the plasma membrane.** To label the majority of the RSV G present during an RSV A2 infection, SBA-488 was delivered in three doses at 12, 18, and 24 hpi to the same

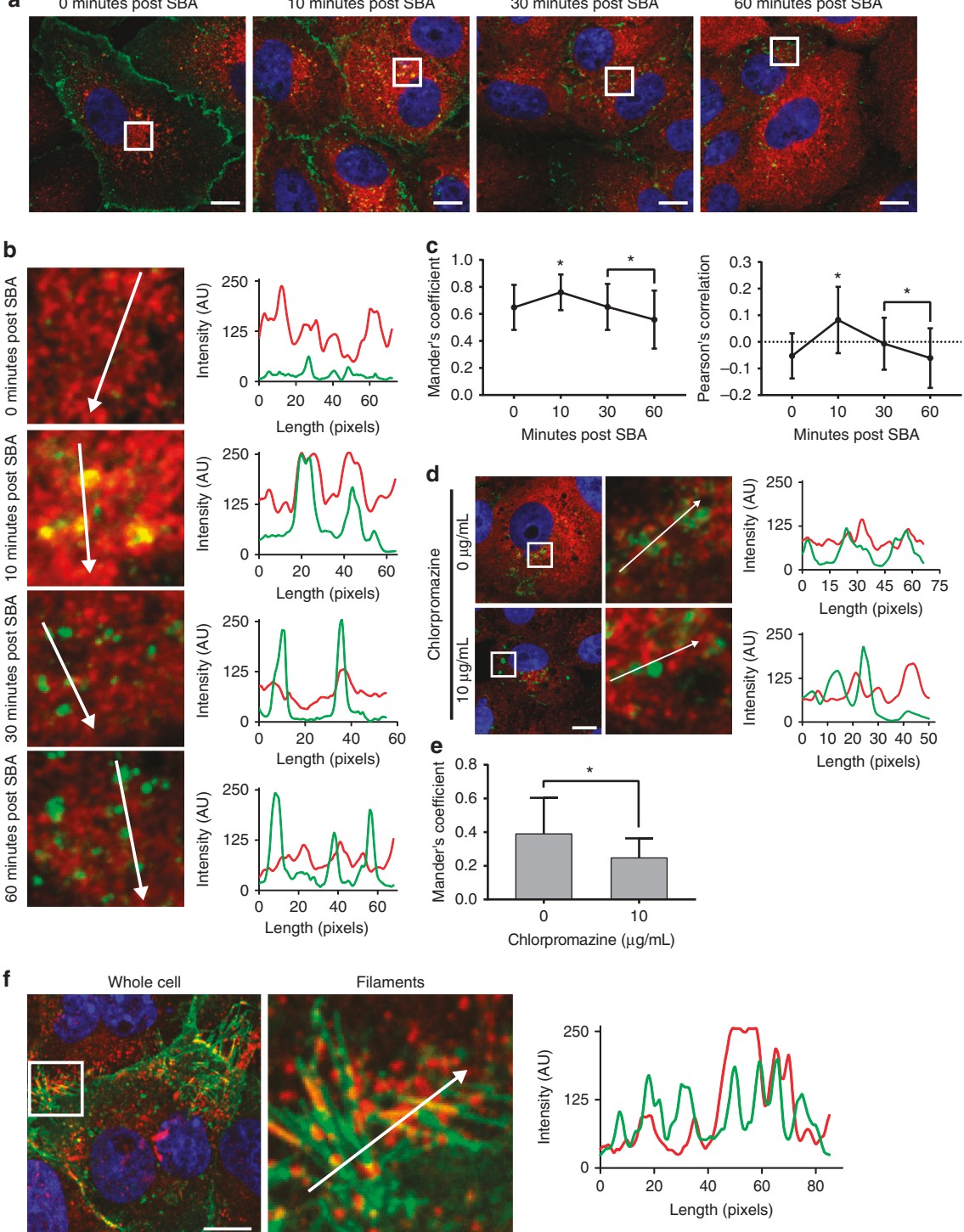

**Fig. 5** RSV G protein internalizes via clathrin and colocalizes with caveolin in viral filaments. **a** RSV infected Vero cells were stained for RSV G with SBA-488 (*green*). The cells were incubated at 37 °C before being fixed and stained for CHC at the indicated time points (*red*). The cell nuclei were stained with DAPI (*blue*). *Scale bar* represents 10 μm. **b** Enlargements of cropped images from the *white boxed regions* in part **a** with intensity profiles showing colocalization along the direction of the *white arrow*. **c** Mander's colocalization coefficient and Pearson's correlation between RSV G and CHC. *Dotted line* indicates 0 correlation. *Error bars* represent standard deviation. *Asterisk* indicates $p < 0.05$ (Mann–Whitney $U$ test). $n = 30$ cells per group. **d** Infected Vero cells at 12 hpi were treated with the indicated concentration of chlorpromazine in media for 2 h. RSV G was labeled with SBA-488 at 4 °C followed by a 1 h incubation at 37 °C in media with drug. Cells were fixed and stained for CHC. *Scale bar* represents 10 μm. Enlargements of cropped images from the *white boxed regions* with intensity profiles showing colocalization along the direction of the white arrow. **e** Mander's colocalization coefficient between RSV G and CHC. *Error bars* represent standard deviation. *Asterisk* indicates $p < 0.05$ (Mann–Whitney $U$ test). $n = 30$ cells per group. **f** Vero cells were infected with RSV A2 at on MOI of 1. At 24 hpi, RSV G was labeled with SBA-488 (*green*) and cells were fixed and stained for caveolin (*red*) and nuclei (*blue*). Enlargement of cropped image from the *white boxed region* with an intensity profile showing colocalization along the direction of the *white arrow*. *Scale bar* represents 10 μm

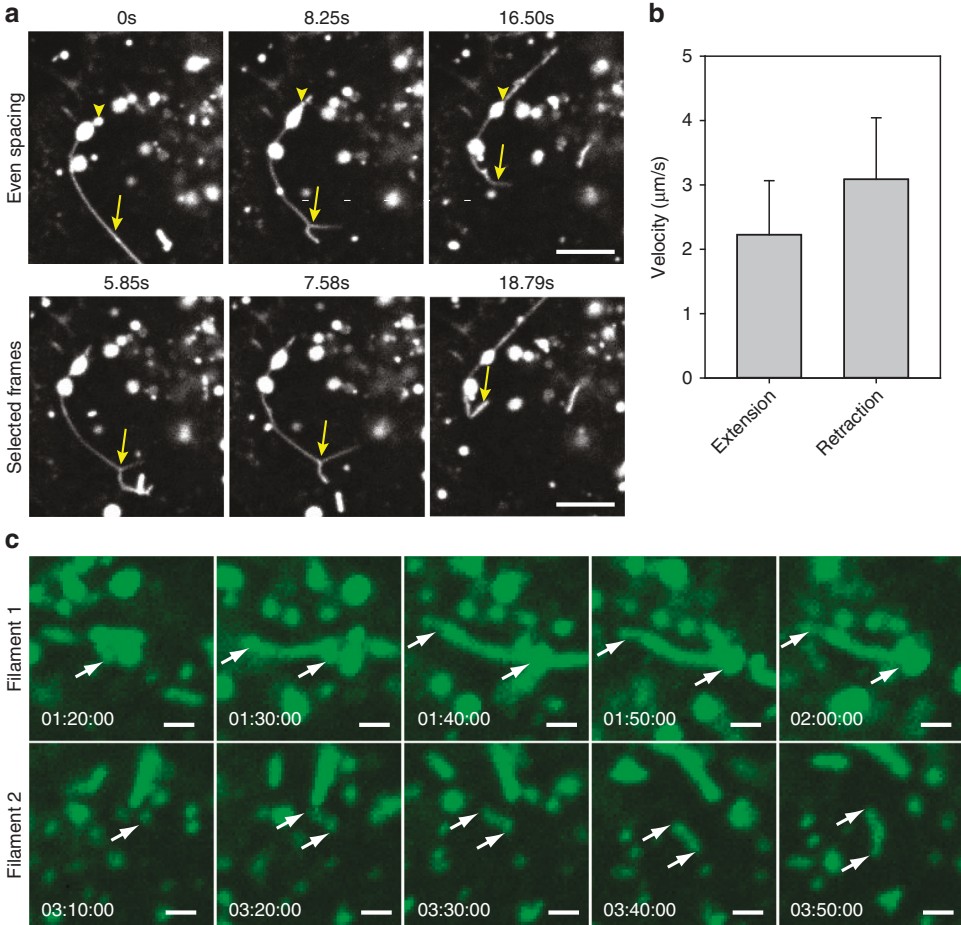

**Fig. 6** RSV filaments are formed by a rapid vesicular extension and retraction motion. **a** Vero cells were infected with RSV A2 at an MOI of 1. At 16 hpi, SBA-488 was delivered and allowed to internalize for an hour. Cells were then imaged live at 12.13 Hz. Time series are shown with either even temporal spacing (*top*) or as selected frames from the movie (*bottom*). *Yellow arrowheads* indicate distension of vesicles, while *arrows* indicate extended filaments and branched filaments. *Scale bar* represents 5 μm. **b** Individual extension and retraction events were analyzed by hand on a frame-by-frame basis to determine the mean velocity. 30 events of each type were analyzed. *Error bars* represent standard deviation. **c** Vero cells were infected with RSV A2 at an MOI of 1. After 12 h, SBA-488 (*green*) was delivered to the cells and allowed to internalize for 1 h before being imaged every 10 min. Cropped regions of two individual filaments are shown. *Yellow arrows mark* the two apparent ends of the filaments. *Scale bars* represent 1 μm

cells before fixation 24 hpi and immunostaining for RSV F. RSV G was found in viral filaments, as expected, as well as in intracellular vesicles, both of which colocalized with RSV F (Fig. 4a). We then performed a similar experiment and delivered an Alexa Fluor 647 conjugated SBA (SBA-647) to infected cells 12 hpi first, and then, SBA-488 at 24 hpi prior to fixation and immunostaining for RSV F (Fig. 4a). As expected, SBA-labeled RSV G was found in viral filaments in both sets of cells, regardless of when it was delivered. RSV G labeled with SBA-647 delivered 12 hpi was found in intracellular vesicles rich in RSV F. Interestingly, RSV G labeled with SBA-488 delivered 24 hpi appeared in vesicles that do not colocalize with either RSV F or SBA-647 labeled RSV G. This finding suggested that RSV G is involved in a complex recycling mechanism where membrane-bound RSV G is first internalized and subsequently incorporated into viral filaments protruding from the surface. To confirm this, we delivered SBA-488 to live RSV A2 infected cells 12 hpi and immediately began imaging every 10 min (Fig. 4b). Compared to the initial time point, we observed a dramatic reduction of SBA-488 staining on the surface of the cell, and a marked increase in the brightness and number of intracellular vesicles over a period of 1 h. This demonstrated that RSV G is rapidly endocytosed into intracellular vesicles. Together, these data indicated that

membrane-bound RSV G recycles from the plasma membrane into intracellular vesicles before being incorporated into viral filaments.

**RSV G-containing vesicles may switch from clathrin to caveolin.** We aimed to determine the mechanism by which RSV G recycles into the cell from the plasma membrane. We delivered SBA-488 to infected cells 12 hpi, fixed the cells 0, 10, 30, and 60 min after delivery, and immunostained for clathrin heavy chain (CHC) (Fig. 5a). We found that RSV G localized to the membrane immediately post-delivery, colocalized highly with CHC 10 min post-delivery and appeared to be released from clathrin-coated vesicles 30–60 min after delivery (Fig. 5b). Both the Pearson's correlation and Mander's coefficient for SBA-488/CHC confirmed these observations (Fig. 5c). This indicated that RSV G is internalized from the membrane through the clathrin endocytic system. We confirmed these results using chlorpromazine, a potent clathrin-mediated endocytosis inhibitor. We incubated RSV infected cells 12 hpi with media containing chlorpromazine for 2 h. We then labeled RSV G with SBA-488 and incubated the cells an additional hour before fixing and staining for CHC (Fig. 5d). In chlorpromazine treated cells, RSV G was found in

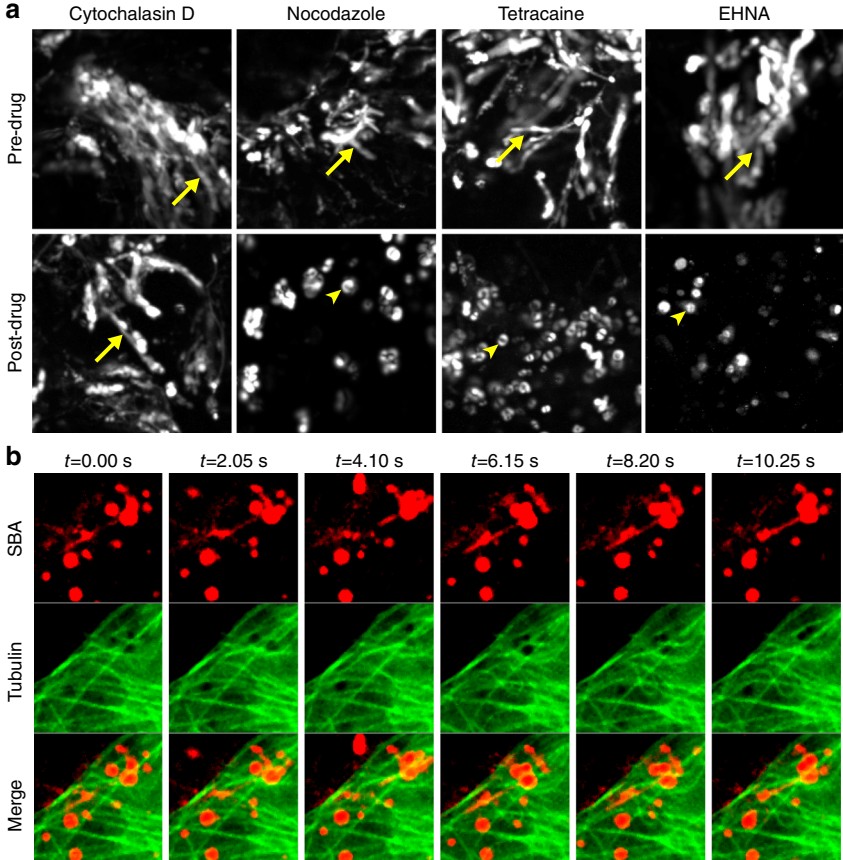

**Fig. 7** Vesicular RSV filament formation is microtubule and dynein dependent. **a** Vero cells were infected with RSV A2 at an MOI of 1. At 12 hpi, SBA-488 was delivered to the cells and allowed to internalize for 1 h. Cells were then imaged live briefly at 9 Hz prior to drug delivery to verify the presence vesicular filament formation. Cells were then treated with nocodazole to disrupt microtubules, cytochalasin D to disrupt actin filaments, tetracaine to inhibit molecular motors, or EHNA to inhibit dynein before being imaged live again. Data is presented as SD maps over 30 s of imaging. *Yellow arrows* indicate fast motion, shown as streaks, while *yellow arrowheads* indicate slow motion, shown as *hollow circles*. Scale bars represent 5 μm. **b** Tubulin-GFP (*green*) was expressed in Vero cells overnight prior to infection with RSV A2 at an MOI of 1. At 12 hpi, SBA-647 (*red*) was delivered and allowed to internalize for 1 h. Cells were then imaged live at 4.9 Hz. *Scale bar* represents 3 μm

large intracellular granules that did not colocalize with CHC (Fig. 5d). These results were confirmed with Mander's colocalization analysis (Fig. 5e). Finally, we labeled RSV G in infected cells 24 hpi before fixing and staining for caveolin and found high colocalization between RSV G and caveolin on viral filaments (Fig. 5f). Together, these data indicated that RSV G recycles into cells via clathrin-mediated endocytosis and over time, colocalizes with caveolin-coated vesicles during the filament formation process.

**RSV filaments are formed via rapid vesicular distension.** Next, we examined the dynamics of filament formation in living cells. We infected cells with RSV A2 and labeled RSV G with SBA-488 16 hpi. After 1 h, cells were imaged live at the maximum speed of our spinning disk confocal system (Fig. 6a; Supplementary Movie 1). Surprisingly, we found that some of the vesicles were distending and retracting into long filament-like structures. Additionally, we noticed that some of these structures appeared to branch, as has been observed previously for RSV filaments[40]. To determine the speed of these events, we measured their length across individual movie frames and found that both the lengthening and shortening of the structures occurred at a speed ≥2 μm/s. To ascertain whether these structures were viral filaments, we fixed the sample immediately after imaging and demonstrated colocalization for RSV F with SBA-488 (Supplementary Fig. 1). To examine the motion of the filaments associated to vesicles, we

infected cells with RSV A2 for 12 h before labeling RSV G with SBA-488. After 1 h, cells were imaged live every 10 min for 12 h (Fig. 6c; Supplementary Fig. 2; Supplementary Movie 2). As expected, we found that RSV filaments extended rapidly from vesicles within a single frame. Furthermore, some of these filaments persisted for a long period of time, suggesting that they had matured from the vesicular extension and retraction phase. These results indicated that RSV filaments are formed by a rapid extension/retraction process of intracellular vesicles before reaching the plasma membrane.

**Microtubules mediate intracellular filament formation.** Because the speed of filament extension and retraction is similar to the previously reported speed of dynein-associated structures, we hypothesized that this motion was governed by molecular motors along microtubules[41]. To investigate this hypothesis, we examined the role of the cytoskeleton and molecular motors in the vesicular extension of RSV filaments. First, we used pharmacological agents to determine which cytoskeletal elements and which molecular motors are involved in the formation of the filaments. The effects of drug treatment were verified with immunofluorescence as previously described[42]. Cells were infected with RSV A2 and stained with SBA-488 12 hpi. After 1 h, cells were imaged live, before and after drug treatment. The motion of single vesicles was analyzed via single particle tracking by computing standard deviation (SD) maps for each image time series

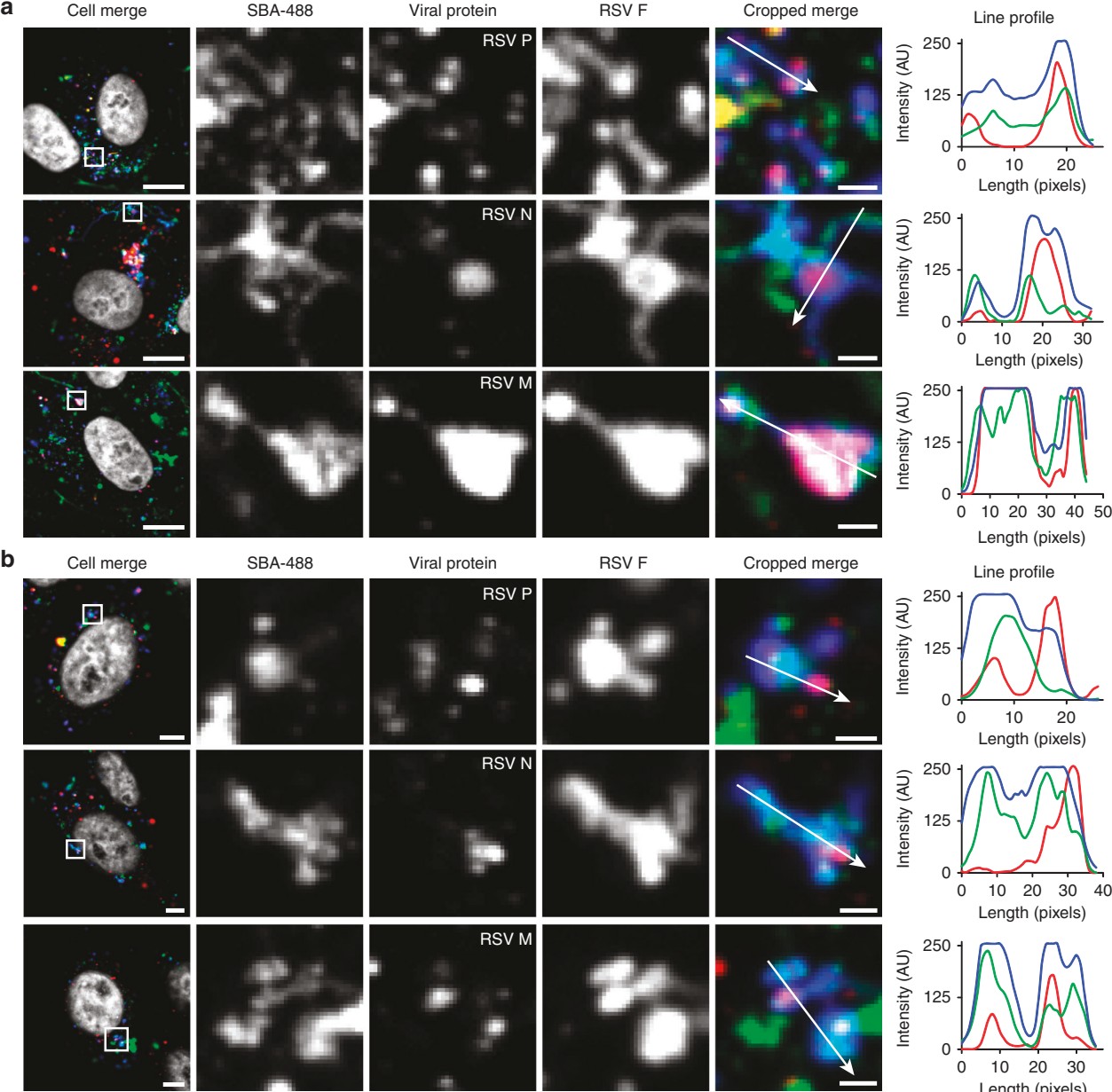

**Fig. 8** RNP granules merge with glycoprotein vesicles prior to the plasma membrane. **a** Vero cells were infected with RSV at an MOI of 1. 12 hpi, SBA-488 (*green*) was delivered and allowed to internalize for 2 h. Cells were then fixed with 1% PFA, permeabilized with 0.1% saponin, and stained for RSV F (*blue*) and either RSV P, RSV N, or RSV M, as indicated (*red*). *Scale bar* represents 10 μm in whole-cell images and 1 μm in cropped images. Enlargement of cropped image from the *white boxed region* with an intensity profile showing colocalization along the direction of the *white arrow*. **b** The same experiment in **a** was repeated in SBA-488 (*green*) stained HEp-2 cells. Cells were stained for RSV F (*blue*) and either RSV P, RSV N, or RSV M as indicated (*red*). *Scale bar* represents 10 μm in whole-cell images and 1 μm in cropped images. Enlargement of cropped image from the *white boxed region* with an intensity profile showing colocalization along the direction of the *white arrow*

over 30 sec. SD maps are generated by calculating the SD of the intensities on a per pixel basis over a given time frame, resulting in a 2D image of time series data. Fast moving particles result in large intensity fluctuations and correspondingly higher values on the SD map[43, 44]. This resulted in fast moving objects appearing as streaks, while slow moving objects will appear as hollow circles, representing diffusive motion over time.

While there was no change in the dynamics of filament extension when cells were exposed to the actin disruption agent cytochalasin D, all vesicular motion was halted when cells were exposed to the microtubule inhibitor nocodazole, to tetracaine, a promiscuous inhibitor of molecular motors, and to erythro-9-(2-hydroxy-3-nonyl)

adenine (EHNA), a dynein-specific inhibitor (Fig. 7a; Supplementary Movies 3 and 4). Unexpectedly, filaments that were present prior to microtubule disruption were static and did not retract. This data suggested that microtubules were involved with the active extension and retraction motion, but not the structural stability of mature RSV filaments. To confirm these results, cells were transduced with GFP-tubulin, infected with RSV, and stained with SBA-647 12 hpi. Extending filaments were found to colocalize or be adjacent to microtubules, further supporting the observation that microtubules are involved with this motion (Fig. 7b). Together, this data indicated that the microtubule motors are necessary for the formation of RSV filaments.

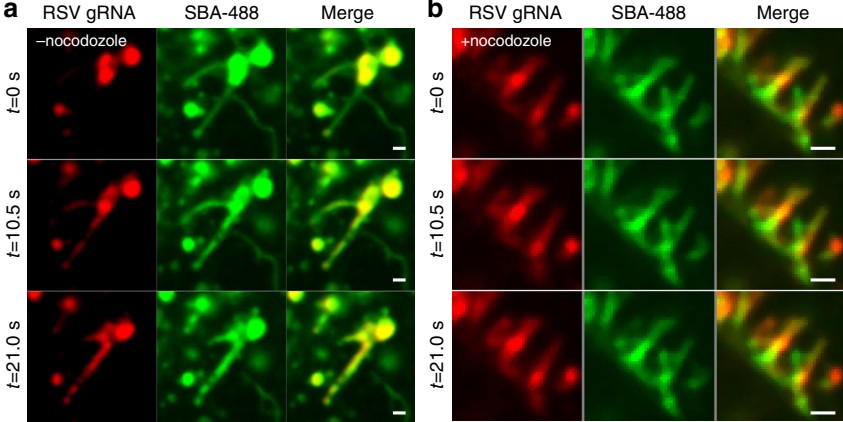

**Fig. 9** RSV genomic RNA is assembled with extended filaments. **a** Vero cells were infected with RSV at an MOI of 1. At 12 hpi, Cy3B-labeled MTRIPs (*red*) targeted to RSV genomic RNA were delivered at 30 nM. After membrane recovery, SBA-488 (*green*) was delivered and allowed to internalize for 1 h. Cells were then imaged live at 4.3 Hz. Images were smoothed with a fine rolling ball filter in Volocity. *Scale bar* represents 1 μm. **b** Vero cells were infected with RSV at an MOI of 1. At 12 hpi, Cy3B-labeled MTRIPs (*red*) targeted to RSV genomic RNA were delivered at 30 nM. After membrane recovery, SBA-488 (*green*) was delivered and allowed to internalize for 1 h. Microtubules were then disrupted using nocodazole. Cells were then imaged live at 4.2 Hz. *Scale bar* represents 1 μm

**RNP granules merge with intracellular glycoprotein vesicles**. Currently, RSV F and RSV G, among other viral proteins, are thought to accumulate on the plasma membrane prior to genomic RNP trafficking, through an unknown mechanism, to the membrane where mature virions assemble[10, 30, 45]. Because we observed that the formation of filaments takes place in the cytosol, as indicated by the involvement of microtubules and motors, we sought to investigate the spatial distribution of RSV M, RSV N, and RSV P relative to the connected vesicles and filaments observed in our live-cell imaging experiments. Since these structures are lost upon permeabilization with 0.2% Triton X-100, we utilized 0.1% saponin to preserve them, as previously described[12]. RSV infected cells were stained with SBA-488 12 hpi and were allowed to incubate for 2 h. Vero cells were then fixed in 1% paraformaldehyde (PFA), and immunostained for RSV F and either RSV M, RSV N, or RSV P (Fig. 8a). In each case, RSV M, RSV N, or RSV P were found predominately in intracellular vesicles, while SBA-488 and RSV F were found both in the vesicles and the attached filaments. To confirm that these results were not limited to Vero cells, the experiment was repeated in HEp-2 cells with similar results (Fig. 8b). These data indicate that RSV M, RSV N, and RSV P are present in vesicles alongside RSV F and RSV G prior to filament formation in the cytosol.

Using RSV N as a marker for viral RNPs, we analyzed the colocalization of SBA-488, RSV N, and RSV F in infected cells, where SBA was delivered immediately prior to fixation (Supplementary Fig. 3a). Since SBA was delivered before fixation and permeabilization, only RSV G on the plasma membrane was labeled. RSV F and RSV N were found to colocalize as early as 6 hpi and remained closely associated at every time point analyzed (Supplementary Fig. 3b). This indicated that viral RNPs and RSV F interact very early during the course of infection within the cytosol, and not only at the membrane as reported previously[19, 30]. This observation was confirmed using thresholded Mander's colocalization analysis (Supplementary Fig. 3c). The percentage of RSV N containing either RSV F or RSV G remained constant during the course of an infection due primarily to the large quantity of cytoplasmic RSV N. The colocalization between RSV N or RSV F and RSV G on the plasma membrane increased only at 24 hpi, indicating that, at this stage, the cytoplasmic granules have merged with the plasma membrane.

**Viral genomic RNA is assembled with extended filaments**. Because viral RNP granules appeared to colocalize with the cytoplasmic RSV G granules at early time points prior to viral filament formation, we hypothesized that viral genomes were assembled with progeny viral filaments after filament extension instead of concurrently. To test this, we used fluorescent, multiply labeled tetravalent RNA imaging probes (MTRIPs) to label and visualize the RSV genome[6, 46, 47]. Cells were infected with RSV A2 for 16 h prior to delivery of MTRIPs via streptolysin O (SLO). After membrane recovery from SLO exposure, SBA-488 was delivered to the cells and allowed to internalize for 1 h before live-cell two-color imaging. Viral genomic RNA initially appeared only in bright SBA-labeled vesicles adjacent to SBA-labeled filaments (Fig. 9a; Supplementary Movie 5). Genomic RNAs were then rapidly localized, within 30 s, from the vesicle to the pre-existing filament, as indicated by the increase of RNA signal along the filament and the corresponding decrease in the associated vesicle. This data supports our hypothesis that RSV genomes are localized to viral filaments after their extension. We then aimed to determine if microtubules were also involved with genomic RNA localization to RSV filaments. We repeated the above experiment upon treatment with nocodazole to disrupt the microtubule network and observed that the motion of all filaments and viral genomic RNAs was halted (Fig. 9b). Together, these data suggest that RSV genomic RNAs are localized to extended filamentous virions via a microtubule dependent mechanism.

## Discussion

To our knowledge, this study is the first instance of using fluorescently conjugated lectins to image RSV glycoprotein dynamics in living cells. We demonstrated that SBA specifically binds RSV G in a manner that does not inhibit the infectivity or replication of the virus, and that SBA permits analysis of RSV G recycling and other dynamics in the context of viral infection.

The apical recycling endosome is critical for RSV infections and specifically for the formation of filamentous virions, but the mechanism involved in the recycling of viral proteins is still unclear[3, 5]. Previous evidence indicated that RSV glycoproteins

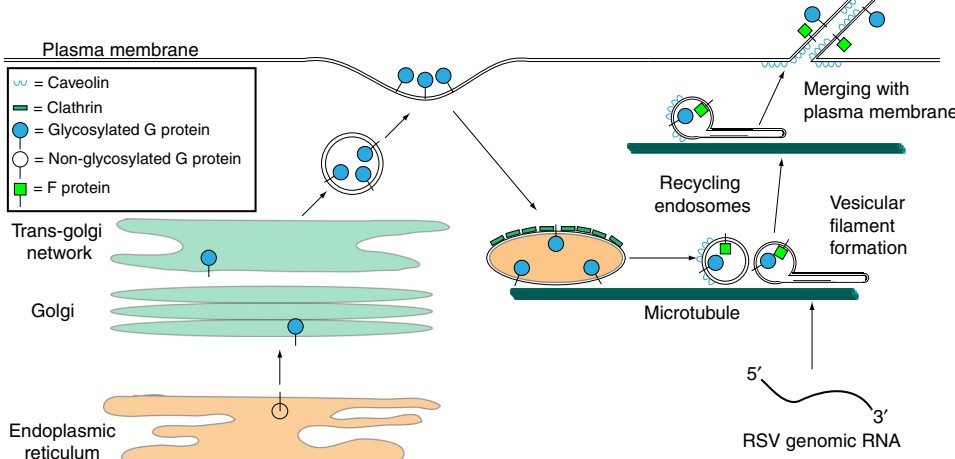

**Fig. 10** Model of RSV filament formation prior to the plasma membrane via the rapid distension of vesicles along microtubules. Progeny RSV G is initially translated into the endoplasmic reticulum before being glycosylated and transported through the golgi network to the plasma membrane. RSV G, labeled with SBA, then is endocytosed and enters the recycling endosome compartment via clathrin-coated vesicles. These vesicles, which also contain RSV F, are distended by dynein along microtubules into filamentous particles prior to being loaded with RSV genomic RNA. During this recycling process, the vesicles colocalize with caveolin. The filaments then mature by merging with the plasma membrane through an unknown process

internalized in host cells via clathrin-mediated endocytosis and that this event is critical for productive RSV infection. However, these were only investigated in the context of viral entry[28, 29]. In this study, we labeled nascent RSV G after translation in the endoplasmic reticulum and transport to the plasma membrane through the Golgi complex. We found that RSV G recycled from the plasma membrane via clathrin-mediated endocytosis and that this mechanism might be associated with the production of RSV filaments, supporting the role of the secretory membrane system in filament formation. Future work will specifically address this issue. Additionally, these findings are substantiated by the recycling of glycoproteins of both Hendra virus and Nipah virus, two other members of the *Paramyxoviridae* family[48, 49]. These results also corroborate previous studies on the role of clathrin-mediated endocytosis of both RSV G and the paramyxovirus parainfluenza virus type 5 (PIV-5) hemagglutinin-neuraminidase glycoprotein[50, 51]. Additionally, our results indicate that RSV G eventually utilizes caveolin-coated vesicles before incorporation into mature virions and filaments, both supporting and elaborating on previous studies on both RSV and PIV-5[30, 31, 52].

RSV filaments play a key role in the infectivity of the virus. Previous results, obtained analyzing fixed cells, were able to identify viral and host-cell proteins necessary in RSV filament formation, but they were unable to investigate the dynamics of filament formation and genome localization as they were occurring[5, 7, 14, 15, 31, 40, 53]. By utilizing SBA-488 to directly label RSV G in living infected cells, we have recorded the first live images of wild-type RSV filament formation. While previous studies have observed filament formation over time, these results were obtained from imaging expressed GFP-fused host-cell factors on the plasma membrane and not by labeling viral proteins[31]. We observed a rapid distension and retraction of vesicular RSV G into viral filaments following its recycling from the membrane. This mechanism is also responsible for the production of branched filaments, which have been observed in previous findings[40]. We measured that the speed of both filament extension and retraction is >2 µm/s, similar to speeds reported for dynein-mediated vesicular transport[41]. These results support earlier reports that microtubules are involved with the assembly of progeny RSV and were confirmed using cytoskeleton disrupting drugs and molecular motor inhibitors[18]. Similarly, the assembly

of filamentous Influenza A was found to be mediated by micro-tubules- and molecular motors-dependent endocytic recycling of vesicles[54–56]. Finally, the role of microtubules in RSV filament assembly indicated that this extension process occurs in the cell cytoplasm prior to the plasma membrane, which supports and expands upon previous findings of RSV filaments budding into intracellular vesicles[11, 12].

Currently, the dynamics or molecular determinants of RSV genomic RNA localization and loading into progeny virions are unclear[45]. In the present study, we used fluorescence microscopy to analyze the colocalization of RNP and glycoprotein granules over the course of an infection and found that RNPs and glyco-protein vesicles merged as early as 6 hpi. Additionally, by using a modified fixation and permeabilization protocol to improve the preservation of the structures seen in our live-cell imaging experiments, we found that these vesicles contained multiple RSV proteins, that are implicated in filament formation, while the filamentous extensions, at early time points, only contained RSV F and RSV G[14]. Because the ectodomain of RSV G appears to be cleaved in Vero cells during recycling, these results were con-firmed in HEp-2 cells[32]. To concurrently track RSV filaments and genome in live infected cells, we utilized SBA-488 to label RSV G and fluorescent MTRIPs to target the RSV genome. We have captured the first recorded images, to our knowledge, of RSV genome loading in filaments, showing that this process occurs after the filament has formed, instead of simultaneously. We also demonstrated that this process is microtubule mediated, indi-cating that it occurs prior to plasma membrane, supporting and expanding on previous reports[11, 12].

Collectively, our results support a new model of RSV assembly and filament production (Fig. 10). We utilized a fluorophore-conjugated SBA to selectively label nascent RSV G in live infected cells without interfering with RSV replication. Using this method, we determined that membrane bound RSV G recycles from the membrane through clathrin-mediated endocytosis. We also sug-gest that during the recycling process, RSV G vesicles undergo a transfer to caveolin-coated vesicles, which is retained in mature filaments. Additionally, we report that RSV filaments and bran-ched virions are formed prior to the plasma membrane through a microtubule and dynein-dependent vesicular extension and retraction motion that occurs during the recycling process.

Finally, we demonstrated that RSV glycoprotein and RNP vesicles merge prior to the plasma membrane and that RSV genomic RNA is loaded into extended filamentous virions in a microtubule mediated manner. While we only describe the use of fluorescent SBA specifically label RSV G, fluorescent lectins can be screened against the glycoproteins of other viruses, possibly allowing for the study of live glycoprotein dynamics in a variety of pathogens. Our data supports a new model in the assembly and loading of RSV filaments, inspiring questions regarding RSV assembly and opening the door to the development of novel therapeutics.

## Methods

**Viruses and cell lines**. Vero green monkey kidney cells (American Tissue Culture Collection CCL-81) or HEp-2 human epithelial cells (ATCC CCL-23) were cultured in DMEM (Lonza) supplemented with 10% fetal bovine serum (FBS) (Hyclone) and 100 U/mL penicillin and 100 mg/mL streptomycin (Life Technologies). Cells were plated on No. 1.5 coverslips (Electron Microscopy Sciences) or 35 mm glass-bottom dishes (In Vitro Scientific) 1 day prior to infection. Both Vero and HEp-2 cell lines were authenticated by ATCC and were checked for Mycoplasma contamination in our laboratory. HEp-2 cells are commonly used to propagate RSV in culture.

Human RSV A2 (ATCC VR-1544) or rgRSV-GFP (a gift from Martin L. Moore's lab, Emory University) was propagated in HEp-2 cells when the cells were >80% confluent. The media was removed and cells were washed with DPBS (without $Ca^{2+}$ and $Mg^{2+}$) from Lonza, and virus was added at a multiplicity of infection (MOI) of 0.1 for 1 h before adding complete medium to the inoculum. Cell-associated virus was harvested by scraping the cells when a high degree of cytopathic effects was visualized (~90% or 96 hpi). Virus was then vortexed briefly, aliquoted, and stored at −80 °C. Virus titers were measured via plaque assay.

**SBA labeling**. SBA conjugated to Alexa Fluor 488 and Alexa Fluor 647 was purchased from Life Technologies and reconstituted, stored, and used per the manufacturer's protocol. Briefly, Vero cells were washed with DPBS without $Ca^{2+}$ and $Mg^{2+}$ before being incubated SBA diluted in HBSS (Lonza) to a final concentration of 20 μg/mL for 10 min unless otherwise specified. For RSV G colocalization and endocytic marker colocalization experiments, cells were chilled on ice for 10 min and SBA was delivered at 4 °C to minimize endocytosis and nonspecific binding. Cells were then washed twice with fresh HBSS before being either fixed, incubated further, or imaged. For live-cell imaging of microtubules alongside SBA, BacMam tubulin-GFP (Life Technologies) was delivered per manufacturer's instructions 16 h before infection with RSV A2 at an MOI of 1.

**Synthetic mRNA production and transfections**. Plasmids for IVT were designed using the full-length nucleotide sequences of RSV F and RSV G from RSV A2 (GenBank M74568.1). The coding region was followed by a 3′ untranslated region derived from the mouse alpha globin sequence. Sequences were codon optimized and inserted in a pMA-7 vector (Thermo Fisher Scientific, GeneArt) to be used as a template for mRNA synthesis. Plasmids were linearized with Not-I HF (New England Biolabs) overnight prior to IVT using a T7 mScript kit (Cellscript) following the manufacturer's instructions. ATP, GTP, and CTP were used alongside m1Y-5′-triphosphate (TriLink). RNAs were capped using 2′-O-Methytransferase followed by enzymatic addition of a poly-A tail, both according to the mScript kit instructions. The capped and tailed mRNAs were then purified using an RNeasy kit (Qiagen), treated with Antarctic Phosphatase for 2 h (New England Biolaboratories), and purified again. Vero cells were transfected using a Neon electroporation system (Invitrogen) into a 24-well plate were transfected with 1 μg of synthetic mRNA encoding RSV F or RSV G, respectively, according to the manufacturer's protocol. 20 h post-transfection, cells were stained with SBA-488 at 4 °C immediately before fixation and immunostaining without permeabilization.

**qRT-PCR**. Vero cells were plated the day before infection in six-well plates. RSV A2 was incubated with 100 μg/mL of either SBA or vehicle control. Cells were then infected with the different virus treatments at an MOI of 1 for 24 h. Total RNA was extracted using an RNeasy kit (Qiagen) following the manufacturer's protocol. mRNA was then isolated from total RNA using an Oligotex kit (Qiagen) following the manufacturer's protocol. cDNA synthesis was performed using the same amount of mRNA per condition and an RT² First Strand kit (Qiagen) following the manufacturer's protocol. qRT-PCR was then performed in triplicate using the RT² SYBR Green ROX qPCR Mastermix (Qiagen) and primer sets (Supplementary Table 1) described in Boukhvalvoa et. al.[57] on a Step-One Plus real-time thermocycler (Applied Biosystems). Rhesus Macaque GAPDH primers (Qiagen) were used as a loading control. Relative quantitation was performed using the Applied Biosystems software. Infections and extractions were repeated twice.

**MTRIP synthesis and delivery**. MTRIP design and synthesis is described in detail elsewhere[46]. Briefly, 2′-O-methyl RNA/DNA chimeric nucleic acid oligonucleotides targeting the RSV A2 genome contain a 5′-biotin modification (Biosearch Technologies) and 5 internal dT amino groups. These ligands were then labeled with either Cy3B NHS ester (GE Healthcare) or DyLight 650 NHS ester (Pierce) according to manufacturer instructions. Unbound dye was removed via centrifugation in a 3 kDa spin column (Millipore) and stored at −20 °C until use. Probes were assembled by mixing labeled oligonucleotides with Neutravidin (Pierce) at a 5:1 molar ration and allowed to react for 1 h at room temperature. Unbound oligonucleotides were filtered out by centrifugation through a 30 kDa spin column (Millipore).

MTRIPs were delivered to cells at 30 nM concentration in Opti-MEM I medium (Invitrogen) containing 0.2 U/mL activated SLO (Sigma) after washing the cells with DPBS without $Ca^{2+}$ and $Mg^{2+}$. The membrane was then allowed to recover in fresh growth medium for 15 min before imaging.

**Antibodies**. For western blot, the primary antibody used was a mouse monoclonal anti-RSV G (Abcam, Cat. No. ab94966) diluted to 2 μg/mL in Odyssey blocking buffer (LI-COR) with 0.1% Tween-20. The secondary antibodies were a donkey anti-mouse IRDye 680RD (LI-COR) and a donkey anti-rabbit IRDye 800 (LI-COR) and were diluted 1:3000 in Odyssey blocking buffer with 0.1% Tween-20.

For immunostaining, primary antibodies used were mouse anti-RSV N (Abcam, Cat. No. ab22501), human anti-RSV F (MedImmune, palivizumab), mouse anti-RSV F (Abcam, Cat. No. ab94968), mouse anti-RSV G (a gift from Ralph Tripp, clone 130-6D), rabbit anti-caveolin-1 (Santa Cruz, Cat. No. sc-894), and mouse anti-clathrin light chain (Biolegend, Cat. No. MMS-423P). The mouse anti-RSV F antibody was used only for the mRNA expression experiments in Fig. 1. All primary antibodies for immunostaining experiments were used at 1 μg/mL. Secondary antibodies used were donkey anti-rabbit Cy3 (Jackson ImmunoResearch), donkey anti-human Alexa Fluor 647, and donkey anti-mouse Alexa Fluor 488 (Life Technologies). All secondary antibodies for immunostaining experiments were used at 4 μg/mL.

**Immunostaining**. Vero cells were fixed with 4% PFA (Electron Microscopy Sciences) for 10 min at room temperature before permeabilization with 0.2% Triton X-100 (Sigma) for 5 min at room temperature. For experiments in Fig. 8, cells were fixed in 1% PFA for 10 min and permeabilized for 15 min in 0.1% saponin. Then, cells were blocked by incubation with 5% bovine serum albumin (Calbiochem) for 30 min at 37 °C before being incubated with primary antibody for 30 min at 37 °C. Cells were then washed with PBS and incubated with secondary antibody for 30 min at 37 °C. Multiple antibody labeling was performed simultaneously after checking cross-reactivity. Cell nuclei were then stained with 4′,6-diamidino-2-phenylindole (DAPI) (Life Technologies), and coverslips were mounted onto glass slides with Prolong Gold (Life Technologies).

**Fixed and live-cell imaging**. Images for long-term infection experiments (Fig. 6b) were recorded with a Hamamatsu Flash 4.0 v2 sCMOS camera on a PerkinElmer UltraView VoX spinning disk confocal microscope mounted to a Nikon Ti-e body with a 60x, NA 1.49 CFI apochromat objective. All live-cell experiments were recorded using cells in 35 mm glass-bottom dishes with Leibovitz's $CO_2$ independent L-15 medium (Life Technologies) supplemented with 10% FBS. Cells and dishes were kept at 37 °C with a Pathology Device LiveCell stage-top incubation system. Images were acquired with the Perfect Focus 3 system engaged to minimize temperature drift.

All other images were acquired with a Hamamatsu Flash 4.0 v2 sCMOS camera on a PerkinElmer UltraView spinning disk confocal microscope mounted to a Zeiss Axiovert 200 M body with a ×63 NA 1.4 plan-apochromat objective. All images were acquired with Volocity (PerkinElmer) with Z-stacks taken in 0.2 μm increments. Cells and dishes were kept at 37 °C during imaging by using a Chamlide TC-L live-cell stage-top environment with objective heater (Live Cell Instrument).

**Drugs**. After MTRIP delivery and SBA-labeling, Vero cells were incubated for 90 min with 1 μM cytochalasin D (Sigma) for actin depolymerization, for 90 min with 4 μM nocodazole (Sigma) for microtubule depolymerization, or for 15 min with 1 mM EHNA (Sigma) for dynein inhibition. After incubation, live-cell imaging was performed in Leibovitz's L-15 media. Tetracaine was used to promiscuously inhibit motors at 100 μM in L-15 and cells were imaged immediately. To inhibit clathrin-mediated endocytosis chlorpromazine (Sigma) was used at 10 μg/mL and was delivered 2 h prior to SBA delivery, and cells were incubated an additional hour in media containing the drug prior to fixation.

**Image processing**. All colocalization and object volume analysis was performed in VolocityMovies were exported from Volocity as TIFF files and imported into ImageJ (NIH) for STD map generation using the ZProject plugin. To obtain intensity profiles, images were exported as TIFF files into ImageJ for analysis using the RGB profiler plugin, and results were plotted in Excel (Microsoft). Linear contrast enhancements were applied to images and videos for clarity. All image quantification and analysis was completed on unenhanced data.

**Western blotting**. Vero cells were plated the day before infection in six-well plates. RSV A2 was incubated with 100 µg/mL of either SBA or vehicle control. Cells were then infected with the different virus treatments at an MOI of 1 for 24 h. Cells were lysed with RIPA buffer (Pierce) containing 1x Complete Protease Inhibitor (Rosche) before being scraped and clarified by centrifugation. Protein concentration was calculated using a BCA assay (Pierce) following the manufacturer's protocol. Lysates were stored at −80 °C until further use.

For gel electrophoresis, 12 µg of lysates were mixed with 4x SDS loading buffer (LI-COR Biosciences), boiled for 10 min at 70 °C, chilled on ice, and loaded into wells of a Novex 10% Bis-Tris precast gel (Life Technologies) alongside a molecular weight marker (LI-COR). Gel was run in an XCell Surlock Mini Cell system (Life Technologies) in 1X MOPS running buffer (Life Technologies) at a constant 200 V for 50 min. Protein was then transferred to 0.45 µm pore nitrocellulose membranes (Life Technologies) in 1× Western transfer buffer (Life Technologies) at a constant 30 V for 1 h using an XCell II Blot module (Life Technologies).

Western blots were stained using a Snap i.d. blot holder (Millipore). Nonspecific binding was blocked using Odyssey blocking solution at room temperature (LI-COR). Vacuum was then immediately applied for 20 s to remove solution. Primary antibody, including a rabbit anti-ß-actin antibody (Cell Signaling Technology, Cat. No. 8457) loading control (diluted 1:1000), was then applied and allowed to incubate for 10 min. Vacuum was then applied and the blots were washed three times with 1x PBS containing 0.1% Tween-20 (PBST). Secondary antibody was then applied and allowed to incubate for 10 min before blots were again washed three times with PBST. Blots were imaged using an Odyssey IR scanner (LI-COR). Only linear contrast enhancements were performed for the final representative images.

**Virus neutralization assay**. rgRSV-GFP stocks were mixed with the indicated concentrations of SBA, palivizumab, or vehicle control and allowed to bind for 2 h on a tube rotator at 37 °C. Vero cells were then infected at an MOI of 1 with the virus treatments or control containing only complete growth medium. 3 hpi, cells were washed with with a 40 mM sodium citrate solution containing 10 mM potassium chloride and 135 mM sodium chloride (Sigma) adjusted to pH 3.0 to eliminate any virus that had not yet internalized before being returned to complete growth medium. 48 hpi, cells were lifted into single-cell suspensions with 0.25% Trypsin (VWR) before being suspended in FACS buffer (DPBS without $Ca^{2+}$ and $Mg^{2+}$, supplemented with 1% FBS and 5 mM EDTA). Cells were then analyzed for GFP fluorescence with an Accuri C6 flow cytometer (BD Biosciences).

**Viral filament isolation**. SBA-488 (20 µg/mL) was mixed with RSV-A2 stocks for 30 min at 37 °C before filtration and isolation of viral filament as described previously[6]. Briefly, SBA-labeled virus was first centrifuged through 5 and 0.45 µm filters (Millipore) at 5000×g and 4 °C for 4 and 1 min, respectively, to remove the dead cell debris from the cell-associated RSV stock. For imaging RSV virion on glass, coverslips were coated with poly-L-lysine (Sigma) in a 24-well plate. Labeled and filtered virion were then centrifuged onto the surface of the coverslips for 30 min at 4 °C and 2400×g. For infecting cells with SBA-labeled virus, titer was adjusted accordingly to an MOI of 1 before inoculating Vero cells.

**Statistical analysis**. Results were plotted and statistical analyses were performed using Prism 7 (GraphPad). Power analysis was performed to ensure adequate sample size for experiments. For two-group comparisons, the Mann–Whitney U test was performed. For comparisons of greater than two groups, one-way ANOVA on ranks or two-way ANOVA was performed as appropriate.

**Data availability**. All data relevant to this study are available from the corresponding authors upon request.

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

## Acknowledgements

This work was funded by NIH grant R01GM094198 and R01GM114561 (P.J.S.) and by a research partnership between Children's Healthcare of Atlanta and the Georgia Institute of Technology (A.W.L.). We would like to acknowledge the Petit Institute Optical Microscopy core facility as well as Nadia Boguslavsky of the Cellular Analysis core facility of the Parker H. Petit Institute for Bioengineering and Bioscience, both at Georgia Tech, for their use of microscopes and the flow cytometer, respectively. We would like to thank Dr. Ralph Tripp for the RSV G primary monoclonal antibody and Dr. Martin Moore for the rgRSV-GFP virus. This material is based upon work supported by the National Science Foundation Graduate Research Fellowship Program under Grant No. DGE-1650044 (E.L.B.). Any opinions, findings, and conclusions or recommendations expressed in this material are those of the author(s) and do not necessarily reflect the views of the National Science Foundation.

## Author contributions

D.V., C.Z., and P.J.S. planned the experiments and interpreted the data. D.V., D.V.S., E.L.B., E.A., and J.L.K. performed all experiments except the long-term imaging experiments. D.V., E.A., and A.W.L. performed the long-term live cell microscopy experiments. D.V., C.Z., and P.J.S. wrote and edited the paper.

## Additional information

**Competing interests:** The authors declare no competing financial interests.

