## [Peer Review File · Nature Communications]

Reviewers' comments:

Reviewer #1 (Remarks to the Author):

Vanover et al present interesting and novel insights into the process of RSV maturation. Using a lectin that binds to RSV G protein, the authors show trafficking of G protein and formation of viral filaments in live cells. The authors identify clathrin as the basis for G protein recycling and show that viral 'filaments' are formed from or at G-containing vesicles and are loaded with RNPs in the cytoplasm in a microtubule-dependent mechanism. The implication of these findings is that virions mature in the cytoplasm and exit by unknown mechanism, rather than form at the plasma membrane. The findings will be of interest to the field as it aims to better understand the life cycle, which will contribute to the development of antiviral concepts. There are some questions as to the claims made by the authors:

One concern is that there are a number of conclusions predicated on SBA specificity for G, while Fig 1 shows reactivity but not specificity. The authors mention in Fig 1a that SBA co-localizes with G and with F. To show that SBA is specific for G, a western of SBA pulldown is incubated with anti-G antibody (Fig 1c). However, to prove specificity (reactivity with G, not F), an anti-F antibody should be included on this western blot. Thus, the extent of SBA binding to F is not addressed. This concern is in connection with previous work (ref 3 cited in this paper), which showed that F uses its C-terminal domain to recycle through the apical recycling endosome. To be confident that the recycling observed in later figures concerns primarily G (and not F), a better demonstration is needed of SBA specificity for G. Specificity for G could also be corroborated by confirming some of the data with G- and F-specific antibodies.

There is also a third RSV transmembrane glycoprotein (SH), with complex glycosylation. SH is not required for filament assembly (neither is G) but is present in viral filaments. It would be prudent to point out the existence of the third glycosylated viral protein as a potential target for SBA.

The concept of cytoplasmic RSV maturation has been presented previously. Using electron microscopy, E. Arslanagic et al, 1996 ('Maturation of respiratory syncytial virus within HEp-2 cell cytoplasm') show the presence of mature RSV particles within cytoplasmic vesicles in HEp2 cells and conclude that maturation of RSV occurs not only at the plasma membrane but also on inner cytoplasmic vesicle membranes. The Arslanagic paper is relevant to the presented work and should be discussed.

Reference 29 (Corry et al) shows that Vero cells are unique in that the majority of the G protein is cleaved by cathepsin L during recycling, and as a result the recycled G is missing a large part of its ectodomain. Could ectodomain removal influence the subsequent trafficking or behavior of G vesicles?

In Fig 2a, the various round particles have different colors in the merge panel. Do the authors have an explanation for this?

In the introduction, the authors state that G has been implicated in maturation and egress. While the role of G in entry has been documented, a role in egress is not clear. Several papers in the field have shown that G is not required for virion production and that infectious virus is generated in full absence of G.

White boxes in Fig 3a are missing.

In Fig. 5D, the authors show caveolin containing vesicles. In the model (Fig. 10), these vesicles are not represented. How do the authors envision the role of the caveolin containing vesicles in the process of filament formation?

The figure legend of Fig 5 is missing section e).

Reviewer #2 (Remarks to the Author):

In this study, Vanover and colleagues developed a new technique to label the respiratory syncytial virus G protein via binding with lectin soybean agglutinin (SBA). Using this reagent, they carefully document G protein as it is endocytosed from the cell surface and subsequently localized to viral filaments by a process that is microtubule dependent.

Endocytosis of paramyxovirus F proteins prior to cathepsin cleavage and packaging into virions has been demonstrated previously for Nipah and Hendra viruses in a pair of papers published in J. Virol. The present study suggests a variation on this theme, in which G protein is endocytosed prior to its assembly on viral filaments.

Although the authors have presented numerous findings that contribute to knowledge in this area, a number of weaknesses were noted:

1. Specificity of the SBA label for viral G protein is not directly demonstrated. Fig. 1C does not demonstrate specificity as stated in the text. It merely shows the presence of RSV G in the SBA-bound fraction. It is not clear whether this binding is specific, or to what extent other proteins besides RSV G were pulled down. Why is there no transfection experiment to rule out any possibility of RSV F being labeled by the SBA?
2. The experiments of Fig. 2 do not convincingly show that SBA-labelled virus has normal infectivity. A kinetic analysis of virus production and protein synthesis (quantified) is needed. In addition, there is significant complication due to the SBA labeling efficiency being less than 100% (as shown in Fig. 2a). The analysis needs to take into account that much of the viral protein seen in Fig. 2c could have come from cells that were infected with the virus particles that did not receive any SBA.
3. The experiments of Fig. 3 show that SBA-labeling of G on infected cells does not affect the amount of F and N fluorescence signal per cell. It is not clear what the point of this is. The concern isn't so much that labeling of G at the cell surface will inhibit synthesis of N or F. The concern is that labeling G at the cell surface will change the characteristics of G. Such

concerns need to be addressed, for example, by showing that infectious virus release over time, the amount of physical particle release, and polypeptide composition of virions, including G incorporation, is not adversely affected in cells that have been SBA labeled.

4. A key conclusion of the paper is that "RSV filaments are formed by a rapid extension/retraction process of intracellular vesicles." (Legend title of Fig. 6). However, it seems that what is being visualized in Fig. 6 is not necessarily the filament formation process, but merely could be the uniting of vesicle-derived G with filaments that have already been recently made through the actions of other proteins such as F, M, P, and N (the minimal requirements for filament formation in transfected cells). Thus, the visuals provided might have nothing to do with the filament formation steps, but rather just illustrate how G is added to the filaments. To track filament formation, it would be important to monitor one of the four proteins that are known to be important for filaments to form.

5. The conclusion of Fig. 8 that "RSV RNP granules merge with RSV G granules prior to localization at the plasma membrane" (Title of Fig. 8 legend; also referred to in the middle of p. 12) is not well supported. These experiments only visualize SBA-labelled G that is on the cell surface. Some evidence is provided for N association with F prior to plasma membrane localization, but there is nothing here to directly support any claims about what G is merging with prior to its localization at the plasma membrane.

6. To demonstrate that findings are not Vero cell-specific, key experiments should be reproduced using a different cell type such as HEp-2.

Reviewer #3 (Remarks to the Author):

Respiratory syncytial virus (RSV) is a very important human pathogen. It is an enveloped virus that buds out of plasma membrane. In this work, the authors labeled the G protein of RSV with fluorescence soybean agglutinin (SBA) and observed that G rapidly recycles from the plasma membrane via clathrin-mediated endocytosis. Taking advantage of their ability to detect RNA genome of RSV in cells, they observed the dynamic formation of filamentous and branched RSV particles, and assembly with genomic ribonucleoproteins (RNPs) and caveolae-associated vesicles that after recycling of G, prior to re-insertion into the plasma membrane. The work was well performed and data convincing. This innovative work provides new knowledge on egress of paramyxoviruses, which includes many important human pathogens besides RSV, especially in filamentous virus formation. That filamentous structure forms before it reaches plasma membrane is groundbreaking. However, there are several questions. The main concern is the choice of G as the protein to illustrate the budding process. As acknowledged by the authors, G is not essential for forming infectious RSV as demonstrated by infectious capability of RSV lacking G, and reports indicating that G is not essential for forming virus particles.

Also, is it possible that only G labeling was observed in Fig.1 due to the fact that G is heavily glycosylated and highly expressed? If the labeling process is not specific for G, can

the observation of G's recycling just be a normal process of membrane protein traffic?

There appeared to have less protein in SBA-lane than the control lane in Fig.2c, which was performed at 24 hours post infection. In Fig.2d, expression level of GFP was measured at 48 hpi. Why different time frames were used?

How do the authors explain the change of topology of G in filament when it is associated with microtubule and when it is at budding site (Fig.10)? Where is the F protein in this model?

Response to reviewers' questions and comments: We have addressed all of the concerns posed by the reviewers as follows:

Reviewer 1:

1. *One concern is that there are a number of conclusions predicated on SBA specificity for G, while Fig 1 shows reactivity but not specificity. The authors mention in Fig 1a that SBA co-localizes with G and with F. To show that SBA is specific for G, a western of SBA pull-down is incubated with anti-G antibody (Fig 1c). However, to prove specificity (reactivity with G, not F), an anti-F antibody should be included on this western blot. Thus, the extent of SBA binding to F is not addressed. This concern is in connection with previous work (ref 3 cited in this paper), which showed that F uses its C-terminal domain to recycle through the apical recycling endosome. To be confident that the recycling observed in later figures concerns primarily G (and not F), a better demonstration is needed of SBA specificity for G. Specificity for G could also be corroborated by confirming some of the data with G- and F-specific antibodies. There is also a third RSV transmembrane glycoprotein (SH), with complex glycosylation. SH is not required for filament assembly (neither is G) but is present in viral filaments. It would be prudent to point out the existence of the third glycosylated viral protein as a potential target for SBA.*

Response to reviewer: We removed the lectin pull-down assay and, we instead expressed individual viral proteins via mRNA transfection, as suggested by reviewer #2. Because RSV F and RSV G bind in solution (see Reference 35), a pull-down assay would result in positive staining for both RSV F and RSV G, even if SBA binds to only one of the proteins. To alleviate this concern, we expressed either RSV F or RSV G in cells individually using *in vitro* transcribed (IVT) modified mRNA (Figure 1c). We then stained the cells live with SBA before fixing and immunostaining for RSV F or RSV G, respectively. We quantified the colocalization and correlation of SBA with either of the proteins and found that SBA colocalized and correlated with RSV G, but not RSV F (Figure 1d). We did not test SH, due to a lack of commercially available reagents to detect the protein.

2. *The concept of cytoplasmic RSV maturation has been presented previously. Using electron microscopy, E. Arslanagic et al, 1996 ('Maturation of respiratory syncytial virus within HEp-2 cell cytoplasm') show the presence of mature RSV particles within cytoplasmic vesicles in HEp2 cells and conclude that maturation of RSV occurs not only at the plasma membrane but also on inner cytoplasmic vesicle membranes. The Arslanagic paper is relevant to the presented work and should be discussed.*

Response to reviewer: We have discussed both the findings of the Arslanagic manuscript (Introduction, 2nd Paragraph) as well as the results of our work in the context of their report (Discussion, paragraph beginning with "Currently, the dynamics....").

3. *Reference 29 (Corry et al) shows that Vero cells are unique in that the majority of the G protein is cleaved by cathepsin L during recycling, and as a result the recycled G is missing a large part of its ectodomain. Could ectodomain removal influence the subsequent trafficking or behavior of G vesicles?*

Response to reviewer: To address the concerns of ectodomain removal in Vero cells, we have repeated a key experiment in HEp-2 cells. We modified the staining protocol to adapt it to Arslanagic *et. al.* This protocol, which utilizes saponin permeabilization, allowed us to observe filaments connected to vesicles similar to those observed in our live-cell imaging experiments. We detected vesicular structures containing RSV F, RSV G, RSV M, RSV N, and RSV P connected to filaments containing RSV F and RSV G. We performed this experiment in both Vero and HEp-2 cell lines (Figure 8a and b) to demonstrate that similar structures and processes were present in both cell types.

4. *In Fig 2a, the various round particles have different colors in the merge panel. Do the authors have an explanation for this?*

Response to reviewer: To address this concern, we explained in the text that RSV harvesting from RSV infected cells, results in a variety of particles after cell scraping, clarification, and filtration. As described by Alonas *et al.*, *ACS Nano*, 2014, these particles can contain varying amounts of viral proteins and nucleic acids. In our work, we observed the same phenomenon – only some of the particles deposited on glass contained all the viral proteins. This issue was clarified in the text relative to figure 2a.

5. *White boxes in Fig 3a are missing. The figure legend of Fig 5 is missing section e).*

Response to reviewer: Missing white boxes and scale bars in Figure 3a, and section e) of figure 5 were added.

6. *In Fig. 5D, the authors show caveolin containing vesicles. In the model (Fig. 10), these vesicles are not represented. How do the authors envision the role of the caveolin containing vesicles in the process of filament formation?*

Response to reviewer: We have updated the model (Fig. 10) to reflect our results for the role of caveolin and as discussed in Ludwig *et. al.*, 2017 (Reference 31).

Reviewer 2:

7. *Specificity of the SBA label for viral G protein is not directly demonstrated. Fig. 1C does not demonstrate specificity as stated in the text. It merely shows the presence of RSV G in the SBA-bound fraction. It is not clear whether this binding is specific, or to what extent other proteins besides RSV G were pulled down. Why is there no transfection experiment to rule out any possibility of RSV F being labeled by the SBA?*

Response to reviewer: We removed the lectin pull-down assay and instead expressed individual viral proteins via mRNA transfection. Because RSV F and RSV G bind in solution (see Reference 35), a pull-down assay would result in

positive staining for both RSV F and RSV G, even if SBA binds to only one of the proteins. To alleviate this concern, we expressed either RSV F or RSV G in cells using *in vitro* transcribed (IVT) modified mRNA (Figure 1c). We then stained the cells live with SBA before fixing and staining for RSV F or RSV G, respectively. We quantified the colocalization and correlation of SBA with either of the proteins and found that SBA colocalizes and correlates with RSV G, but not RSV F (Figure 1d). We did not test SH, due to a lack of commercially available reagents to detect the protein.

8. *The experiments of Fig. 2 do not convincingly show that SBA-labelled virus has normal infectivity. A kinetic analysis of virus production and protein synthesis (quantified) is needed. In addition, there is significant complication due to the SBA labeling efficiency being less than 100% (as shown in Fig. 2a). The analysis needs to take into account that much of the viral protein seen in Fig. 2c could have come from cells that were infected with the virus particles that did not receive any SBA.*

Response to reviewer: To address this concern, we infected cells with SBA-labelled virus and performed both western blots and qRT-PCR for both viral proteins and mRNAs at various time points to assess the effect of SBA on the amount of viral proteins and mRNAs produced by the infection (Figures 2c, d, and f). Densitometry analysis of the western blots and relative quantification of the qRT—PCR were compared by two-way ANOVA to the results obtained using unlabeled virus. No difference was observed between SBA-labeled and unlabeled RSV at any time point for any protein or mRNA.

9. *The experiments of Fig. 3 show that SBA-labeling of G on infected cells does not affect the amount of F and N fluorescence signal per cell. It is not clear what the point of this is. The concern isn't so much that labeling of G at the cell surface will inhibit synthesis of N or F. The concern is that labeling G at the cell surface will change the characteristics of G. Such concerns need to be addressed, for example, by showing that infectious virus release over time, the amount of physical particle release, and polypeptide composition of virions, including G incorporation, is not adversely affected in cells that have been SBA labeled.*

Response to reviewer: To address these concerns, we performed the following assays:

- 1) **To measure the effect of SBA labeling on RSV G incorporation into virions, we infected cells with RSV A2 and labeled RSV G with SBA-488 12 hpi. 24 hpi, we isolated filamentous virions, deposited them onto glass and stained for RSV G and RSV F and compared the sum intensity of RSV G per filament. These results are described in Figure 3c and d. We found no statistically significant difference in the amount of RSV G on filamentous particles collected from labeled and unlabeled infected cells.**
- 2) **To measure the effect of SBA staining on infectious virus release, we infected cells with RSV A2 and labeled RSV G with SBA 12 hpi. 24 hpi, we**

assayed the supernatants for viral titer by plaque assay and found no statistical difference between SBA at 12 hpi and vehicle control by Mann-Whitney U test. These results are summarized in Figure 3e.

10. *A key conclusion of the paper is that “RSV filaments are formed by a rapid extension/retraction process of intracellular vesicles.” (Legend title of Fig. 6). However, it seems that what is being visualized in Fig. 6 is not necessarily the filament formation process, but merely could be the uniting of vesicle-derived G with filaments that have already been recently made through the actions of other proteins such as F, M, P, and N (the minimal requirements for filament formation in transfected cells). Thus, the visuals provided might have nothing to do with the filament formation steps, but rather just illustrate how G is added to the filaments. To track filament formation, it would be important to monitor one of the four proteins that are known to be important for filaments to form.*

Response to reviewer: To address this concern we developed a staining protocol similar to Arslanagic *et. al.* and used saponin permeabilization to allow visualization of the connected vesicle and filament structures similar to the ones observed in our live cell imaging experiments (Figure 8). At 12 hpi, infected cells were stained with SBA 488, which was allowed to internalize for 2 hours. Cells were then fixed, permeabilized with saponin, and stained for RSV F and either RSV N, RSV M, or RSV P. Both vesicles and filaments contained RSV G (marked by SBA) and RSV F, while in many cells the vesicles only, and not the filaments, contained RSV N, RSV M, and RSV P (while still connected to the filaments). In cells that produced more viral proteins, or were further along in the infection life-cycle, N, M, and P were found in the filaments (data not shown).

This supports our live cell imaging conclusions, that vesicular dynamics described in Fig. 6 is responsible for viral filament formation, and not only for the loading of RSV G onto filaments.

11. *The conclusion of Fig. 8 that “RSV RNP granules merge with RSV G granules prior to localization at the plasma membrane” (Title of Fig. 8 legend; also referred to in the middle of p. 12) is not well supported. These experiments only visualize SBA-labelled G that is on the cell surface. Some evidence is provided for N association with F prior to plasma membrane localization, but there is nothing here to directly support any claims about what G is merging with prior to its localization at the plasma membrane.*

Response to reviewer: We refer to the new version of Figure 8 to address these concerns. In addition to RSV N, we included staining for RSV P and RSV M 12 hpi, to demonstrate that these proteins are associated with RSV G and RSV F vesicles prior to the membrane. We did not stain for M2-1 or L, as the antibodies for these two proteins have high background when staining cells.

12. *To demonstrate that findings are not Vero cell-specific, key experiments should be reproduced using a different cell type such as HEp-2.*

Response to reviewer: The experiment was repeated in Vero and HEp-2 cells (Figure 8a and b) with similar results.

Reviewer 3:

13. *The main concern is the choice of G as the protein to illustrate the budding process. As acknowledged by the authors, G is not essential for forming infectious RSV as demonstrated by infectious capability of RSV lacking G, and reports indicating that G is not essential for forming virus particles.*

Response to reviewer: We merely used RSV G as a marker for filamentous virions. While not required for progeny virion production, RSV G is highly incorporated into filamentous virions, rendering it an ideal “marker” to elucidate the formation of filaments in live cells.

14. *Also, is it possible that only G labeling was observed in Fig.1 due to the fact that G is heavily glycosylated and highly expressed? If the labeling process is not specific for G, can the observation of G's recycling just be a normal process of membrane protein traffic?*

Response to reviewer: We included additional experiments to demonstrate that SBA specifically binds the G protein (Figure 1c and d). We indeed chose RSV G as a target because it is highly expressed and heavily glycosylated. The heavy glycosylation of the RSV G protein allows fluorescently labeled-SBA to bind the glycoprotein, and thus be visible via fluorescent microscopy in both live and fixed cells.

It should be noted that SBA binds to RSV G specific glycans and NOT directly to the protein itself. This is especially advantageous, because SBA does not significantly interfere with RSV infection (Figure 3) and viral processes. Also, we do think that the recycling of the G is a normal part of its trafficking, but that this process is important and essential for filament formation. This is supported by our observations that they are endocytosed via clathrin mediated endocytosis, and are a part of filament formation within the context of caveolae. We feel that this recycling is critical, as the labeled G on the surface at 12 hpi consistently localized to filaments on the cell surface at 24 hpi.

15. *There appeared to have less protein in SBA-lane than the control lane in Fig.2c, which was performed at 24 hours post infection. In Fig.2d, expression level of GFP was measured at 48 hpi. Why different time frames were used?*

Response to reviewer: The western blot for RSV G was repeated in duplicate with replicate wells in each run to demonstrate that there was indeed little difference between SBA-labeled and control virus (Figure 2c).

GFP expressing cells were quantified at 48 hours instead of 24 hours due to the reduced permissivity of rgRSV-GFP compared to RSV A2 (now figure 2e). We allowed the virus to proceed an additional day to account for this. This point was addressed in the associated text in the results section.

16. How do the authors explain the change of topology of G in filament when it is associated with microtubule and when it is at budding site (Fig.10)? Where is the F protein in this model?

Response to reviewer: Based upon the results of our current experiments, delineating the location of RSV F relative to the intracellular vesicles and filaments (Figure 8), we added RSV F to the cartoon in Figure 10 and modified captions and text accordingly.

Currently our data supports the model that molecular motors and the microtubules mediate the vesicular filament formation observed in the movies. The molecules that directly mediate the merging of these structures and the plasma membrane at the budding site is unclear, and will be the focus of future work.

Reviewers' comments:

Reviewer #1 (Remarks to the Author):

The authors have addressed many of the concerns of the previous submission, and the work provides very interesting data useful in the RSV field with regard to understanding how RSV particles form. One weakness with data based entirely on fluorescent images, is that it's difficult for the reviewer to gauge how strong a phenotype is. Nevertheless the paper is very well executed. Two concerns remain:

1) SBA specificity for G (Fig. 1).

Fig 1d coefficients suggest that SBA does not bind F. However, in Fig 1C, it is not clear why cells were permeabilized with triton prior to incubating with anti-F antibodies. From the F staining pattern and the clear visibility of the nucleus in panel 'Expressed RSV protein', it appears that the staining of F is largely cell-internal (as opposed to the same panel with G where the nucleus is not visible due to the strong surface staining of G). It appears there may be little F on the surface, and internal F protein would not be available for SBA staining done live in the first step. The data therefore do not convincingly exclude SBA reactivity for F. Some reactivity with F would not compromise all data but could subtly affect some of the claims. More convincing in Fig. 1 would be data that show F expressed on the cell surface and not reacting with SBA.

2) The link with caveolin (Figures 4 and 5).

G internalizes following clathrin machinery, but co-localization with F in granules is not obvious (Fig. 4). G and caveolin are co-present in filaments at 24 hpi (Fig 5f) but how each protein gets there is not entirely clear. The authors claim a 'switch' for G from clathrin to caveolin vesicles but the evidence for that is lacking. Is clathrin-internalization of G a 'deliberate' step toward filament incorporation, or could clathrin-internalization and filament incorporation be separate (independent) processes (with G internalizing in part due to its strong surface presence and following the cell's clathrin machinery). This concern could be addressed by 'suggesting' a link between G-vesicles and caveolin vesicles as a potential mechanism rather than claiming such a link.

Reviewer #2 (Remarks to the Author):

This manuscript has been extensively modified and improved. My concerns have all been sufficiently addressed in the revised version.

Response to reviewers' questions and comments: We have addressed all of the concerns posed by the reviewers as follows:

Reviewer 1:

1. *SBA specificity for G (Fig. 1).*

Fig 1d coefficients suggest that SBA does not bind F. However, in Fig 1C, it is not clear why cells were permeabilized with triton prior to incubating with anti-F antibodies. From the F staining pattern and the clear visibility of the nucleus in panel 'Expressed RSV protein', it appears that the staining of F is largely cell-internal (as opposed to the same panel with G where the nucleus is not visible due to the strong surface staining of G). It appears there may be little F on the surface, and internal F protein would not be available for SBA staining done live in the first step. The data therefore do not convincingly exclude SBA reactivity for F. Some reactivity with F would not compromise all data but could subtly affect some of the claims. More convincing in Fig. 1 would be data that show F expressed on the cell surface and not reacting with SBA.

Response to reviewer: To address the concern that RSV F was not on the plasma membrane of the transfected cells, we repeated the experiment in which we first transfect the cells with either RSV G or RSV F encoding mRNA before labeling the cells with SBA, fixing, and immunostaining for RSV F or RSV G. This time, we removed the permeabilization step so that the RSV G or RSV F staining would be predominately surface bound (Fig. 1C). Note that the RSV F staining now appears to be on the surface of the cells. We updated the colocalization measurements with the new data (Fig. 1D) demonstrating a higher degree of colocalization of SBA with G and no colocalization with F. Additionally, we added cropped cross-sectional views (YZ plane) of the cells from Fig. 1C along with line profiles across the cell (Fig. 1E). These data clearly show that RSV G and RSV F are present in a layer above the nucleus of the cell, on the plasma membrane.

2. *The link with caveolin (Figures 4 and 5).*

G internalizes following clathrin machinery, but co-localization with F in granules is not obvious (Fig. 4). G and caveolin are co-present in filaments at 24 hpi (Fig 5f) but how each protein gets there is not entirely clear. The authors claim a 'switch' for G from clathrin to caveolin vesicles but the evidence for that is lacking. Is clathrin-internalization of G a 'deliberate' step toward filament incorporation, or could clathrin-internalization and filament incorporation be separate (independent) processes (with G internalizing in part due to its strong surface presence and following the cell's clathrin machinery). This concern could be addressed by 'suggesting' a link between G-vesicles and caveolin vesicles as a potential mechanism rather than claiming such a link.

We have updated the relevant text in multiple locations to reflect that we observe clathrin-internalization and later colocalization with caveolin vesicles, and we suggest that this change might be implicated in the production of filaments rather the definitively stating such (Discussion, Paragraph starting with "The apical recycling endosome...."). We also note that in future work we will address this question specifically.

REVIEWERS' COMMENTS:

Reviewer #1 (Remarks to the Author):

In the revised manuscript, the authors have adequately addressed all remaining concerns.